# Recent Progress of Electrochemical Energy Devices: Metal Oxide–Carbon Nanocomposites as Materials for Next-Generation Chemical Storage for Renewable Energy

**Dohyeong Seok †, Yohan Jeong †, Kyoungho Han, Do Young Yoon and Hiesang Sohn \***

Department of Chemical Engineering, Kwangwoon University, Seoul 01897, Korea

\* Correspondence: hsohn@kw.ac.kr or sonisang@gmail.com

† These authors contributed equally to this work.

**Abstract:** With the importance of sustainable energy, resources, and environmental issues, interest in metal oxides increased significantly during the past several years owing to their high theoretical capacity and promising use as electrode materials for electrochemical energy devices. However, the low electrical conductivity of metal oxides and their structural instability during cycling can degrade the battery performance. To solve this problem, studies on carbon/metal-oxide composites were carried out. In this review, we comprehensively discuss the characteristics (chemical, physical, electrical, and structural properties) of such composites by categorizing the structure of carbon in different dimensions and discuss their application toward electrochemical energy devices. In particular, one-, two-, and three-dimensional (1D, 2D, and 3D) carbon bring about numerous advantages to a carbon/metal-oxide composite owing to the unique characteristics of each dimension.

**Keywords:** metal oxide; carbon; composite; energy storage; electrochemical device

---

## 1. Introduction

Studies on renewable energy storage devices and use are an emerging issue owing to increases in energy consumption and a reduced supply of fossil fuels [1–3]. Electrochemical energy devices including lithium (or sodium) ion batteries and supercapacitors received attention owing to their vast applications, from electronic devices to electric vehicles or energy storage systems (ESSs).

Intensive studies were conducted on the development of high-performance lithium/sodium ion-based electrochemical energy devices because of their high theoretical capacity (~3862 mAh/g for lithium ion-based batteries, and 1165 mAh/g for sodium ion-based batteries). However, despite the tremendous amount of progress made regarding electrochemical energy devices during the last 20 years, they are not suitable for application in a high-performance ESSs or electric vehicles requiring an elongated stability or large battery capacity [4–7]. More specifically, lithium-ion batteries use graphite as an anode material, which is currently available commercially [8–13]. Lithium ions are intercalated/deintercalated during the charge/discharge process to achieve a stable electrochemical reaction, but they have a relatively lower theoretical capacity of 372 mAh/g [14–17]. In sodium-ion batteries, however, it is difficult to apply the same graphite ($d_{002}$ = 0.334 nm) used in lithium-ion batteries as a negative electrode material owing to the large size of sodium ions [18]. Also, commercialized supercapacitors using carbon-based materials have problems with low energy density [19,20].

It is logical to pursue novel electrode materials demonstrating their application in electrochemical devices with an improved performance. Among the many capacitive candidate materials, metal oxides received significant attention owing to their high electron storage capability and emissivity

---

through a chemical redox reaction in an energy storage device [21–23]. However, a metal-oxide-based electrode material undergoes severe volume changes during the charge/discharge process (e.g., lithiation/delithiation and sodiation/desodiation) owing to the conversion reactions that occur with the guest ions (Li and Na) [24,25].

In this context, there were tremendous efforts made toward improving the performance of energy devices using metal oxides and addressing the low-capacity problems of carbon-based materials. If metal oxides are applied, the theoretical capacity of the carbon-based materials used in lithium and sodium ion batteries will be increased [26–31].

However, owing to the large volume change in the active metal-oxide material generated during the battery charging/discharging process, such material is separated from the electrode, thereby reducing the electrical contact, hindering the long-term cycle stability. In addition, owing to the low electrical conductivity of metal oxides, the charge transfer resistance of the electrode is high, which is unfortunate in terms of the output density and rate characteristics of the battery [32,33].

To solve the problems of low electric conductivity and large volume change of the active material during the charging/discharging process of the energy device, which are disadvantages of metal oxides, studies were carried out on the formation of a composite with carbon material. As summarized in Figure 1, synergistic effects can be achieved based on the high theoretical capacity of metal oxides, as well as the high electrical conductivity and capacity for accommodating a good volume change of the carbon-based material, thereby improving the performance of the energy device by forming a carbonaceous material-based metal-oxide composite [34–40].

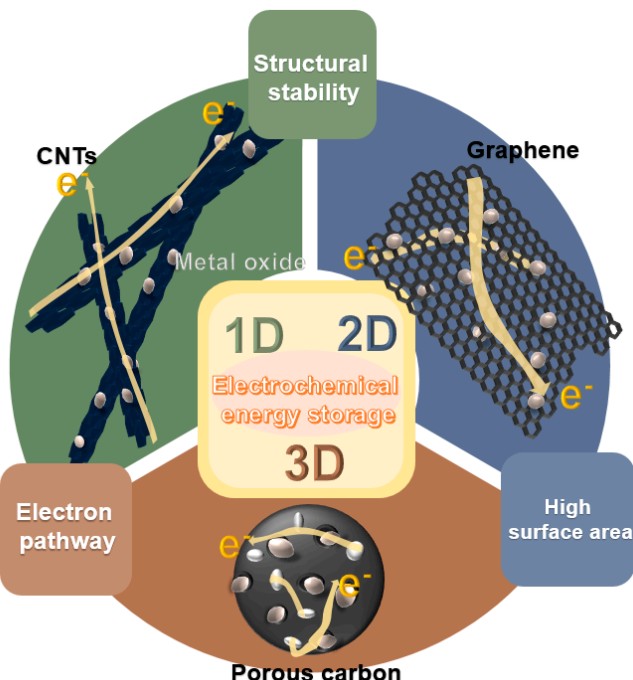

**Figure 1.** Schematic illustration of carbon/metal-oxide composites of various dimensions for electrochemical energy devices.

In this review, we introduce studies on improving the performance of energy devices through the combination of metal oxides and one-, two-, or three-dimensional (1D, 2D, or 3D) carbon materials. The properties of such materials and composites with metal oxides formed on them are also introduced. In addition, we discuss research applied to energy devices based on the properties of these composites.

## 2. Carbon/Metal-Oxide Composite Materials

As mentioned, metal oxides have a high electron storage capacity as electrode materials for electrochemical energy devices, but have a problem of low electrical conductivity. To solve these issues, numerous studies were conducted on improving the electrochemical performance of metal oxides through the formation of a composite material.

There are many different composites made of carbon and metal oxide that were applied to electrochemical energy devices. In terms of their structure, they can be divided into 1D–3D structured composites.

### 2.1. 1D Structured Carbon/Metal-Oxide Composite

One-dimensional structured carbon/metal-oxide composites were extensively studied owing to their unique composite nanotube/fiber structure combined with metal oxide. As a typical architecture, 1D materials of carbon nanotubes (CNTs) or carbon nanofibers (CNFs) typically form a composite structure through a homogeneous decoration of metal oxides around them. As-formed composites exhibit enhanced characteristics coming from their unique structure. Specifically, CNTs with a high aspect ratio in a continuous conductive network can facilitate the charge transport because of the reduced contact resistance with adjacent nanoparticles superior to that of a pristine metal-oxide nanostructure. That is, CNTs can serve as a "facilitated electron transport path or electron highway", which allows a charge transport along the longitude direction [41].

Considering that the conductivity of the electrode material in an energy storage device plays an important role in the power density, CNTs clearly attracted significant attention as co-electrode materials by improving the rate capability of electrochemical energy devices through their excellent charge transport properties [42].

As summarized in Figure 2, the formation of a CNT/metal-oxide composite has several advantages. Firstly, the low electrical conductivity of the metal oxide can be compensated for by the CNTs, which provide a continuous network for electron transport as an "electronic highway" [43]. CNTs can also achieve high electrical conductivity through less mass loading compared to round-shaped nanoparticles by forming a percolation network, which enables a high energy density to be achieved in energy storage devices. Secondly, the high specific surface area of the CNTs increases the contact area between the electrolyte and the electrode. Thirdly, CNTs with an enhanced mechanical toughness compared with pure metal oxide can act as a carbon scaffold through chemical and physical bonding with metal oxides [44].

Zhao et al. compared the electrical conductivities of $Co_3O_4$ with $Co@Co_3O_4$/CNTs at 18 MPa to investigate the effect of CNT compositing on the material properties. A $Co@Co_3O_4$/CNT nanocomposite was synthesized through an arc discharge and low-temperature oxidation (Figure 3a) [45]. Despite the low original conductivity of $Co_3O_4$ ($7.1 \times 10^{-4}$ S/m), the electrical conductivity of the $Co@Co_3O_4$/CNTs was improved by $10^4$ to 7.6 S/m, suggesting the important role of CNTs as an electronic conductive highway (Figure 3b). TEM images (Figure 3c,d) clearly show the consistent formation of a conduction path for electron transport in $Co@Co_3O_4$/CNT nanocomposites through the formation of CNT conductive networks throughout the $Co@Co_3O_4$ nanoparticles (Figure 3e). It is noteworthy that a $Co@Co_3O_4$/CNT nanocomposite exhibits approximately 10,000-fold higher electrical conductivity than that of pristine $Co_3O_4$ without CNTs. Such an enhanced electrical conductivity of a metal-oxide/CNT composite enables a facilitated charge transport, leading to an improved electrochemical performance of energy storage devices based on a composite electrode [46].

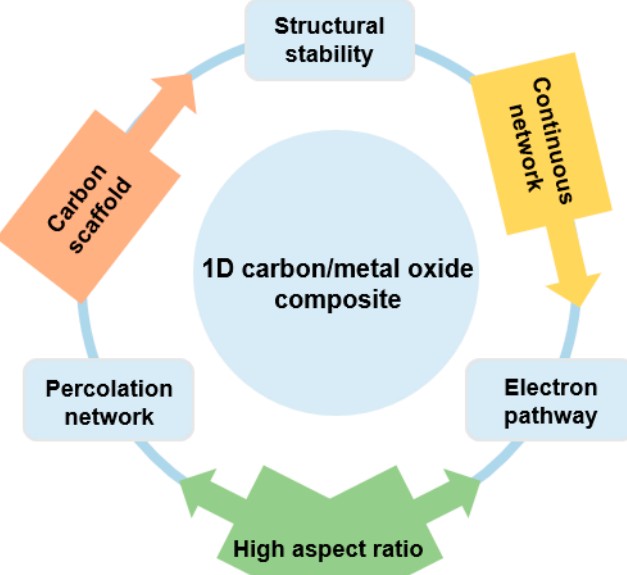

**Figure 2.** Overview of one-dimensional (1D) carbon/metal-oxide composite for energy storage device.

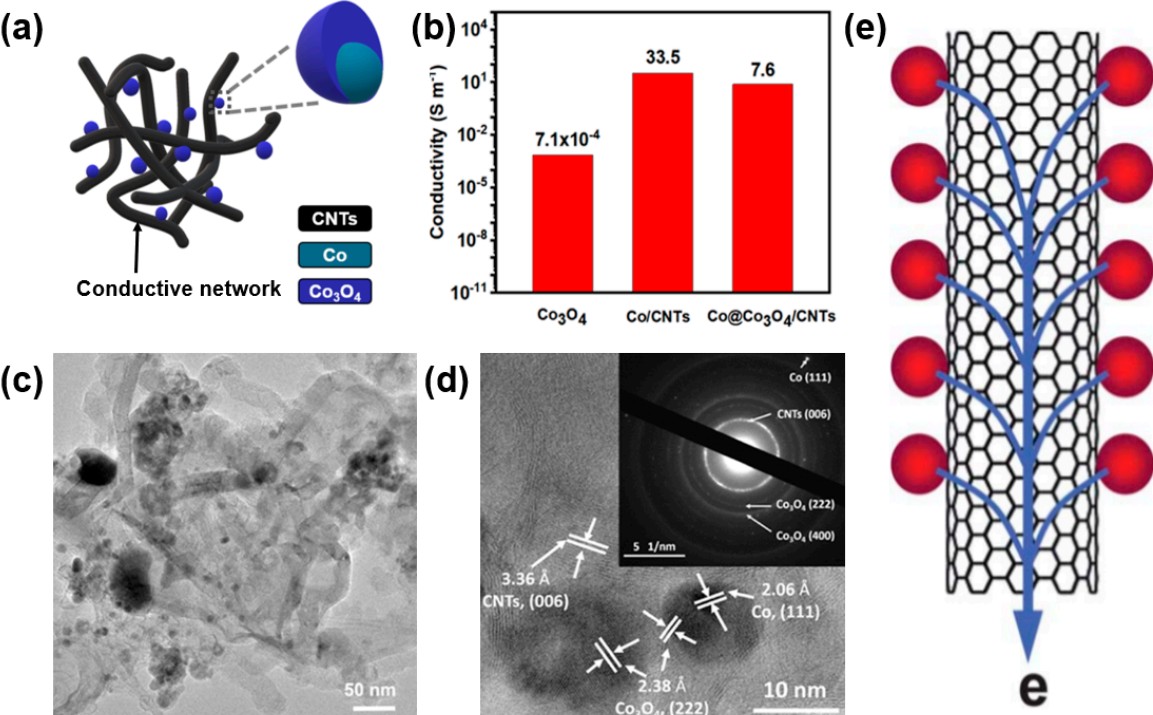

**Figure 3.** (**a**) Scheme of Co@Co$_3$O$_4$/carbon-nanotube (CNT) composite made up of Co$_3$O$_4$ nanoparticles with CNT networks. (**b**) Enhanced conductivity of Co@Co$_3$O$_4$/CNT composite through the formation of a composite with CNT networks. (**c**) TEM and (**d**) selected area electron diffraction (SAED) images of Co@Co$_3$O$_4$@CNTs. Reprinted with permission from Reference [45]; copyright 2018 American Chemical Society. (**e**) Electronic transport characteristics of a CNT/metal-oxide composite. Reprinted with permission from Reference [43]; copyright 2010 Royal Society of Chemistry.

Reportedly, the volume changes of the electrode material used in electrochemical energy devices (i.e., lithium-ion battery) during the charge/discharge process induce cracks or voids through the applied mechanical strain [47]. It is, therefore, logical to alleviate the mechanical stress on an electrode material to improve the electrochemical performances (i.e., cycle stability). As mentioned previously,

intensive studies were conducted on reducing the mechanical strain caused by a volume change of metal-oxide-based electrode materials (Figure 4a,b).

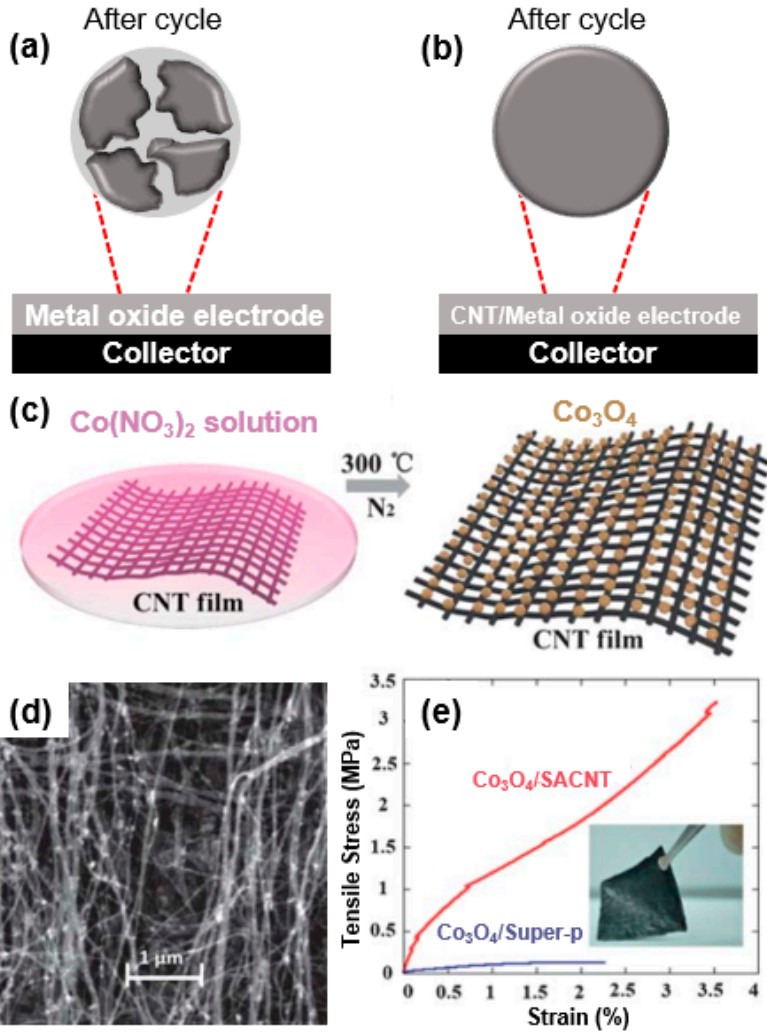

**Figure 4.** Scheme of morphological change of metal oxide based electrode: (**a**) Cracks and voids formed on metal oxide by volume change-induced mechanical stress during cycling. (**b**) Retained morphology of CNT/metal-oxide-based electrode after cycling. (**c**) Scheme of $Co_3O_4$/super-aligned CNT (SACNT) composite in nano-sized $Co_3O_4$ particles grown on SACNT film. (**d**) SEM images of $Co_3O_4$/SACNT. (**e**) Stress–strain curves of $Co_3O_4$/SACNT and $Co_3O_4$/Super-P composites; the inset is a digital photograph of a $Co_3O_4$/SACNT composite. Reprinted with permission from Reference [48]; copyright 2013 Royal Society of Chemistry.

For instance, He et al. prepared $Co_3O_4$/super-aligned CNT (SACNT) composites ($Co_3O_4$/SACNT) in which $Co_3O_4$ nanoparticles were grown homogeneously on super-aligned CNTs through a pyrolysis method (Figure 4c) to reduce the effect of the mechanical strain occurring during a volume change. As-formed $Co_3O_4$/SACNT composites exhibited a high Young's modulus (160 Mpa) and tensile strength (3.5 MPa) and strain (3.2%) at break (Figure 4c,d) [48]. The mechanical stability of a $Co_3O_4$/SACNT electrode is 6.4-fold higher than that of $Co_3O_4$/Super-P, and the tensile strength and strain at break are 27.2- and 1.6-fold better. Overall, a $Co_3O_4$/SACNT composite exhibits superior mechanical strength (higher strength and flexibility) to that of a simple mixture composite ($Co_3O_4$/Super-P) (Figure 4e). Such high flexibility and strength of a $Co_3O_4$/SACNT composite can be attributed to the high mechanical properties of the SACNT, allowing it to withstand the volumetric changes during the charge/discharge process of the energy storage device, leading to stable cycle performance.

By contrast, in a CNT/metal-oxide composite, the ratio of CNT to metal oxide plays an important role in determining the performance of an electrochemical energy device. As mentioned, there was tremendous effort to enhance the electrical conductivity of metal oxides to improve the performance (energy density) of such devices [49]. Such an enhancement was mainly carried out through the compositing of metal oxide with various nanostructured carbon-based conductors including carbon black, acetylene black, carbon nanotube, and graphene [50]. Because conductive nanostructured carbon facilitates charge transport from the redox sites present in the metal oxide, several studies showed an improved electrochemical performance when applied to an energy device [44].

To make full use of this approach, it is necessary to have the appropriate fraction of metal oxide in the capacitive composite material (CNT/metal oxide). In this respect, as shown in Figure 5a, 1D nanomaterials with high aspect ratios should be used as a conductive filler in the percolation network of the composite, enabling facilitated electron motion in the shortest conductive pathway.

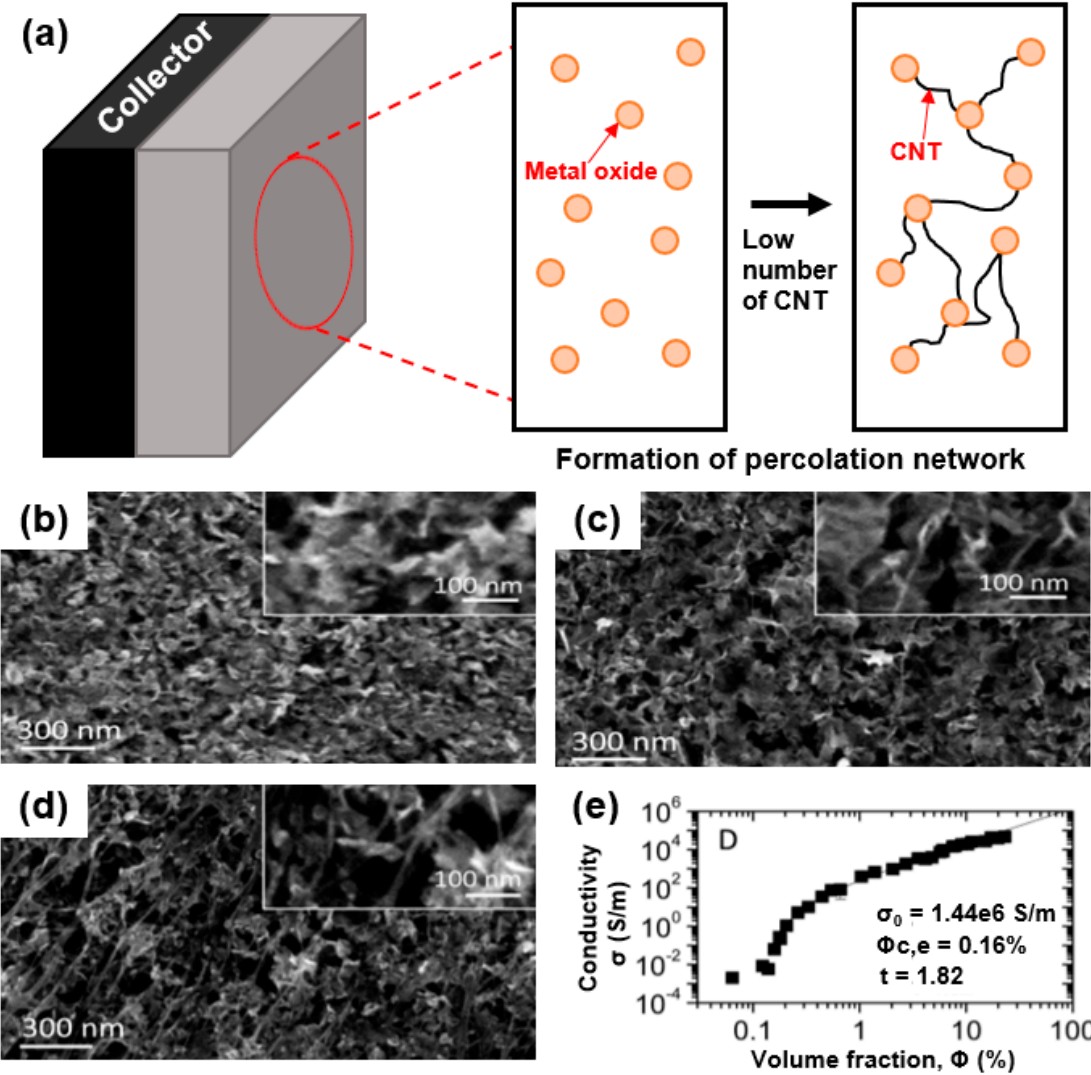

**Figure 5.** (**a**) Formation of percolation network in electrode material with 1D nanomaterial-based composite. (**b**) SEM images of $MnO_2$, (**c**) $MnO_2$/single-walled CNT (SWNT) composite with 1 wt.% SWNT, and (**d**) $MnO_2$/SWNT composite with 25 wt.% SWNT. (**e**) Percolation characteristics of 1D nanostructured CNTs: in-plane electrical conductivity of $MnO_2$/SWNT plotted versus SWNT volume fraction. Reprinted with permission from Reference [50]; copyright 2014 American Chemical Society.

As revealed by Higgins et al., the electrochemical performance of a supercapacitor can be significantly improved using a composite electrode of CNTs and a $MnO_2$ nanoplate [50]. As shown

in Figure 5b–d, 2D structured $MnO_2$ (Figure 5b) hybridized with 1D structured CNTs (1 wt.% CNT (Figure 5c), 2.5 wt.% CNT (Figure 5d)) exhibits multi-dimensional structures. To maximize the electrochemical performance of the capacitive material ($MnO_2$), it is necessary to optimize the CNT amount through the formation of a smooth conductive pathway in the $CNT/MnO_2$ composite. Based on the percolation theory, the CNT content is represented by the volume fraction ($\varphi$) ($\varphi = V_{NT}/(V_{NT} + V_{MnO2})$). As shown in Figure 5e, the electrical conductivity increased tremendously to $10^5$ S/m at $\varphi$ = 25 vol.% of the CNTs, whereas an electrical conductivity of ~100 S/m occurred with an extremely small number of CNTs (1 vol.%) in the composite. Because the electrical percolation of CNTs can be satisfied beyond the threshold value of the nanoconductor (CNTs) in the volume fraction ($\varphi$), the critical volume fraction ($\varphi_{c,e}$) should be achieved in the framework, such as a nanoconductor/insulator composite. As shown in Equation (1), an extremely low electrical conductivity can be exhibited through the percolation theory [50].

$$\sigma = \sigma_0(\varphi - \varphi_{c,e}) \qquad (1)$$

where $\sigma_0$ indicates the conductivity of the nanoconductor, and $n$ is the percolation exponent.

As shown by Higgins et al., an extremely small value of $\varphi_{c,e}$ = 0.16 vol.% can be obtained as a result of the fitting, where $\sigma_0$ is $1.4 \times 10^6$ S/m and the percolation exponent is 1.82. This low percolation threshold can be attributed to the unique composite structure of 1D structured CNTs with large aspect ratios hybridized with metal oxide. Based on this percolation theory, a substantial conductive pathway can be introduced without significantly reducing the amount of capacitive material (metal oxide) in the composite owing to the high electrical conductivity of a CNT network [44].

In a similar way, when used as an electrode in a storage system, the electrochemical properties of nanoconductor/insulator composites such as $CNT/MnO_2$ can be effectively improved using the percolation theory to overcome the low electrical conductivity of $MnO_2$. Specifically, the formation of a composite of metal oxide and CNTs results in an excellent theoretical capacity through an electrochemical redox reaction. This result suggests an effective improvement in the electrochemical performance of $MnO_2$ through the formation of a composite with a conductive agent. In addition, 1D structured CNTs can further improve the performance because of the excellent mechanical properties and charge transfer characteristics of a CNT network. Such formation of a continuous conductive percolation network can lead to an improved performance of the energy storage device without a noticeable debilitation in the capacity during the charge/discharge process [51].

### 2.2. 2D Structured Carbon/Metal-Oxide Composite

A 2D structured carbon/metal-oxide composite requires the use of a carbon nanosheet. As a representative carbonaceous 2D nanosheet, graphene has a honeycomb structure of $sp^2$-hybridized carbon atoms. Owing to the unique 2D structure, graphene has numerous advantages when used to enhance the performance of an energy device (Figure 6) [52]. Firstly, graphene facilitates the electrochemical kinetics owing to its high electrical conductivity and intrinsic carrier mobility. Secondly, because graphene has an extremely large surface area (theoretical surface area of 2620 $m^2$/g), as well as a very large surface-to-volume ratio, it can provide many active reaction sites and effectively shorten the ion diffusion distance. Thirdly, graphene has a high Young's modulus (1.0 TPa) and tensile strength, which improves the device stability and cycle stability. Despite the many advantages of graphene, as an electrode material, it does not achieve a good performance (or it exhibits rapid performance deterioration during the cycling) owing to its re-stacking during the electrochemical reactions [53].

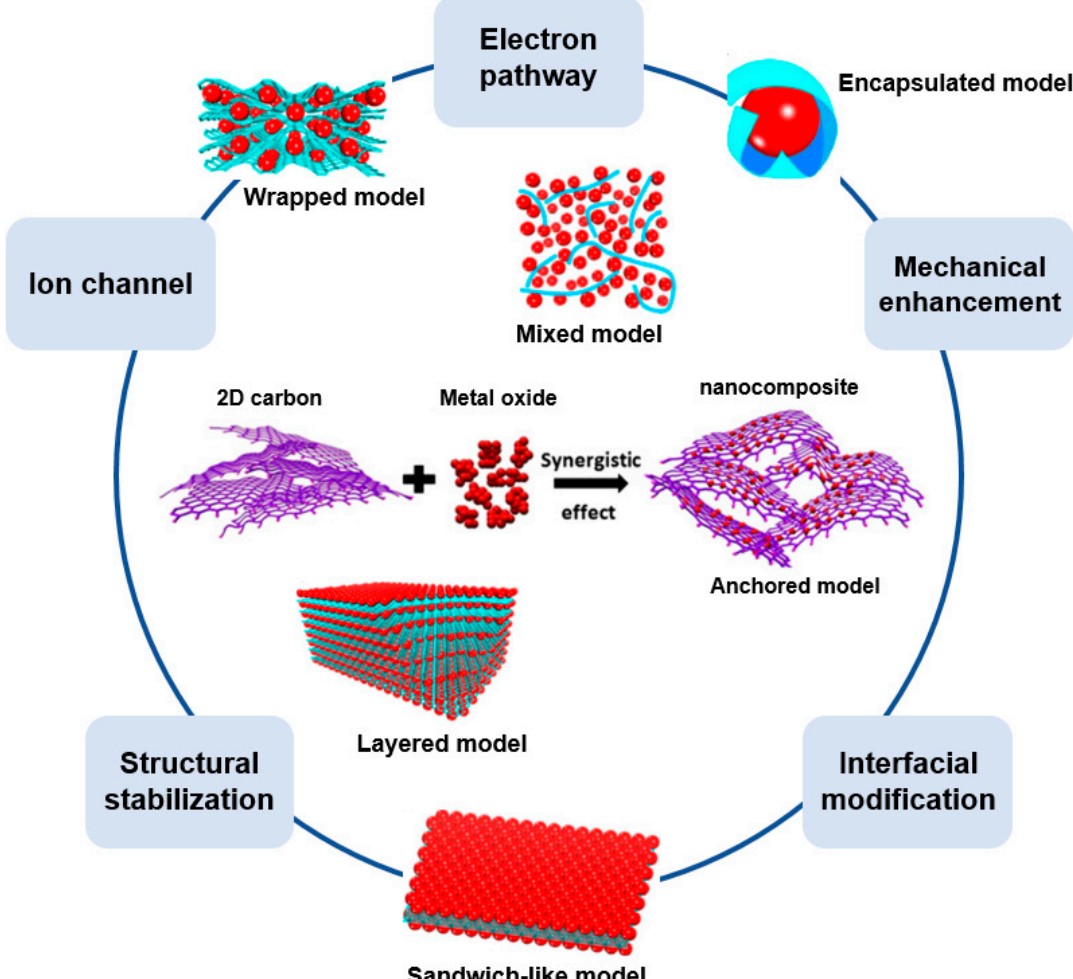

**Figure 6.** Advantages of graphene/metal-oxide composite in various structures. Adapted with permission from Reference [53]; copyright 2011 Elsevier.

In this context, there were intensive studies on composites of graphene and metal oxide because the advantages of graphene and the benefits of metal oxide as a capacitive material can be achieved simultaneously when used as an electrode in an electrochemical energy device [54].

As described in the previous section, metal oxide used as an electrode material of an electrochemical energy device has limited applications in commercial energy devices owing to performance deterioration from low electrical and ionic conductivity and severe volume changes, inducing a mechanical deformation of the metal oxide during the charge/discharge process [55]. There were many efforts to alleviate the above problems through the compositing of a capacitive material (metal oxide) with graphene because it can enhance the conductivity of capacitive materials, as well as mechanically accommodate the volume change-induced strain occurring during the charge/discharge. As described above, such an enhanced performance of a composite can be attributed to the unique material characteristics of graphene. Specifically, graphene with a modified surface can enhance the structural stability of a metal-oxide-based composite owing to its chemical stability and compatibility with metal oxide, thereby creating physical and chemical connection sites and achieving a high electrochemical performance (high capacity and long cycle stability).

As a common form of a graphene/metal-oxide composite, nanostructured metal oxide dispersed or anchored to a graphene nanosheet is highlighted in Figure 7a [56]. For instance, Su et al. synthesized a graphene/$Fe_3O_4$ (GN-$Fe_3O_4$) nanocomposite of $Fe_3O_4$ nanoparticles (7 nm) uniformly grown on a graphene sheet (Figure 7b) [57]. As shown in the TEM images, no $Fe_3O_4$ nanoparticles can be observed

in an aggregated form or found outside a GN sheet (Figure 7b,c) owing to the strong interaction between the metal oxide and graphene.

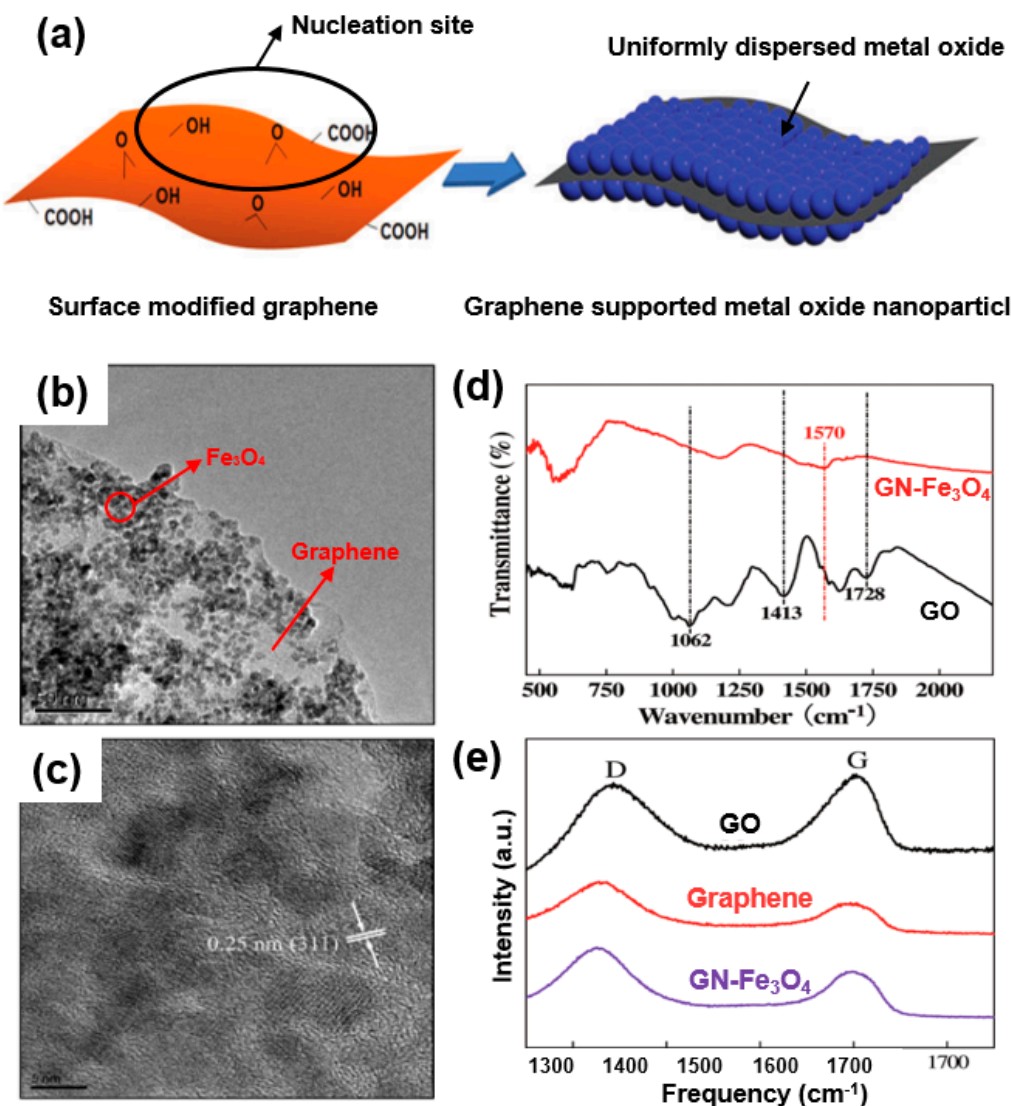

**Figure 7.** (**a**) Metal-oxide formation on surface-modified graphene. Reprinted with permission from Reference [56]; copyright 2012 Wiley-VCH. (**b**) High-magnification TEM images of graphene/$Fe_3O_4$ (GN-$Fe_3O_4$). (**c**) High-resolution (HR)-TEM image of $Fe_3O_4$ on graphene. (**d**) Fourier-transform infrared (FT-IR) spectra of graphene oxide (GO) and GN-$Fe_3O_4$. (**e**) Raman spectra of GO (black), GN (red), and GN-$Fe_3O_4$ nanocomposites (purple). Reprinted with permission from Reference [57]; copyright 2011 American Chemical Society.

The graphene oxide (GO) used in the composite was further analyzed through Fourier-transform infrared (FT-IR) spectroscopy (Figure 7d), the results of which suggest the formation of many functional groups CO ($1413\,\mathrm{cm^{-1}}$) and tertiary C–OH groups ($1728\,\mathrm{cm^{-1}}$). However, differing from the observation of oxygen-containing groups in GO, no peaks corresponding to all oxygen-containing groups were observed after the synthesis of GN-$Fe_3O_4$. Raman spectra (Figure 7e) applied to analyze the carbon characteristics of the composite indicate that the G-band of GO shifts to a lower wavenumber in GN-$Fe_3O_4$. In comparative spectra of GO, graphene, and a composite, the characteristic carbon spectra (G- and D-bands) of GO exhibit a partial shift toward a lower wavenumber in the GN-$Fe_3O_4$ composite, suggesting the structural retention of the graphene/$Fe_3O_4$ composite after long cycles of the charge/discharge process in electrochemical energy devices. Such structural stability of a composite is

attributed to the strong interactions of $Fe_3O_4$ and graphene as a conductive network. Chemical analyses (FT-IR spectra (Figure 7d) and Raman spectra (Figure 7e)) indicate that the GN-$Fe_3O_4$ composite maintains structural and cyclic safety owing to the strong interactions of $Fe_3O_4$ and graphene. This indicates that the well-reduced graphene can also act as a conductive network.

In a similar vein, metal oxide wrapped with graphene, such as in a wrapped model, encapsulated model, or layered model, received considerable attention owing to its excellent device performance [53]. For instance, Zhou et al. synthesized a graphene-wrapped $Fe_3O_4$ (GNS/$Fe_3O_4$) composite as an anode for a lithium-ion battery (Figure 8) [58]. The SEM images and schemes of the synthesized composites are shown in Figure 8a,b. This type of graphene/metal oxide allows the graphene to have an open porous system, and the porous texture of the graphene makes it possible to achieve a flexible electrode because there is no tight connection to other adjacent nanosheets. The metal oxide confined in this flexible porous structure is controlled not only by the porous graphene matrix but also by the metal oxide present between the graphene layers, thereby suppressing the re-stacking of the graphene. Taking full advantage of this, Zhou et al. further compared the particle size of $Fe_3O_4$ with that of commercial $Fe_3O_4$ in a GNS/$Fe_3O_4$ composite before and after 30 cycles to confirm the structural advantages described above. Commercial $Fe_3O_4$ showed an average particle size of 735 nm at the initial stage and 428 nm after the cycling. In contrast, the GNS/$Fe_3O_4$ composite showed an initial average particle size of 196 nm (Figure 8c), whereas the sizes of composite particles showed limited growth after the cycling owing to the structural stability of the composite achieved by the wrapped graphene surrounding the $Fe_3O_4$ (Figure 8d). That is, the average size of the GNS/$Fe_3O_4$ composite was found to be similar to the average size after cycling owing to the unique graphene-wrapped structure of the GNS/$Fe_3O_4$ composite [58].

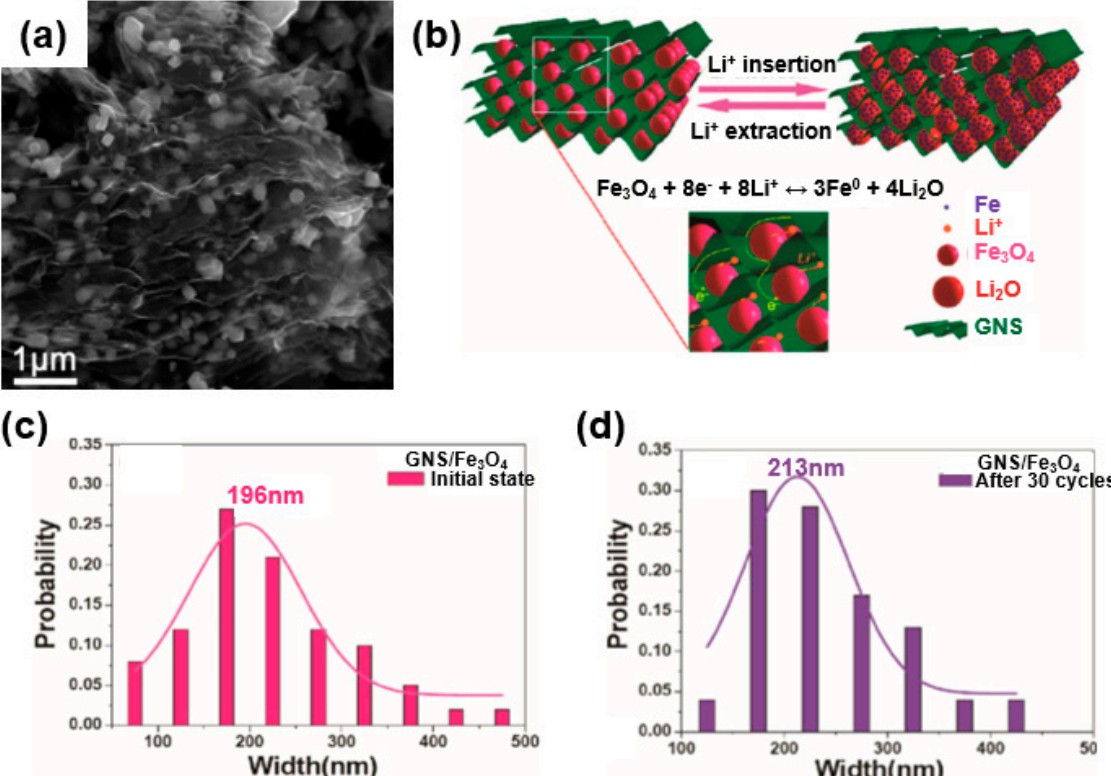

**Figure 8.** Wrapped model: (**a**) SEM image of graphene-wrapped $Fe_3O_4$ (GNS/$Fe_3O_4$) composite. (**b**) Mechanical stress accommodation and ion and electron transport mechanism owing to the structural advantages of the graphene-wrapped model during an electrochemical reaction. (**c**) Size distributions of the GNS/$Fe_3O_4$ composite during the initial state and (**d**) after 30 cycles. Reprinted with permission from Reference [58]; copyright 2010 American Chemical Society.

By contrast, a novel type of graphene (holey graphene) as a 2D structured porous material received increased attention owing to its many advantages when used as a material in an electrochemical energy device [59].

Firstly, as described in Figure 9a, holey graphene has a high surface area, which can provide more active sites and enable the effective wetting and penetration of the electrolyte through the pore channel owing to the porous structure [60]. In addition, porous graphene with interconnected structures can provide a continuous charge transfer pathway and short ion transport pathway during an electrochemical reaction [61]. Note that non-porous graphene also achieves ion diffusion near the edge or nanosheet junction, although the length of the ion diffusion has a longer transport path than direct ion diffusion occurring in porous graphene. When forming a 2D porous graphene and metal-oxide composite for electrochemical energy storage, the composite has the advantages of a reduced path length, large surface area, and facilitated electrolyte penetration, and it can effectively accommodate the strains from the volume change of metal oxide during the charge/discharge process owing to the highly porous structure [59].

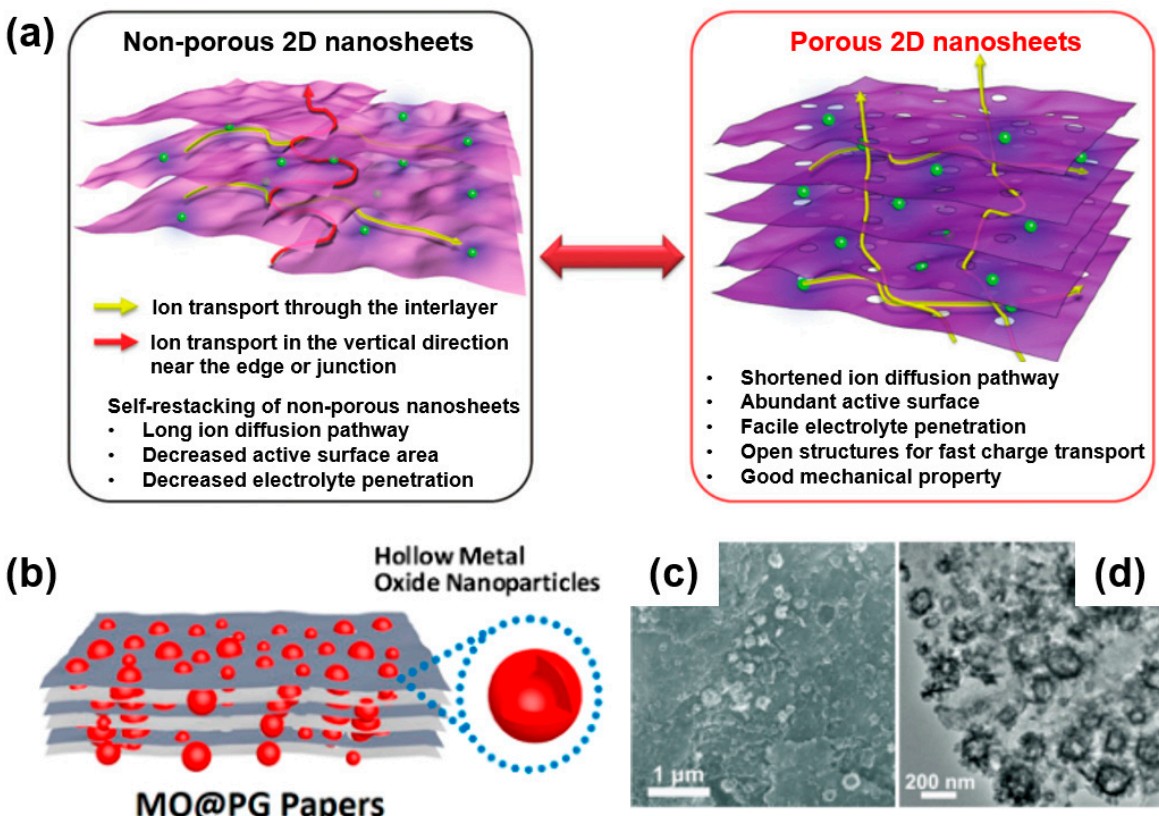

**Figure 9.** (**a**) Schematic of advantageous features of porous two-dimensional (2D) structured material for electrochemical energy device. Reprinted with permission from Reference [53]; copyright 2018 Wiley-VCH. (**b**) Schematic image of porous graphene and metal-oxide composite (MO@PG papers). (**c**) SEM and (**d**) TEM images of MO@PG composite after 4000 cycles at 5 A/g. Reprinted with permission from Reference [62]; copyright 2018 American Chemical Society.

For instance, Zhang's group demonstrated a composite of metal oxide and porous graphene (MO@PG) to improve the electrochemical performance as a cathode material of a lithium-ion battery (Figure 9b) [62]. As shown in SEM (Figure 9c) and TEM (Figure 9d) images, the MO@PG nanocomposite of MO is embedded in the pores of porous carbon rather than the basal plane of the graphene. In addition, as graphene forms a conductive network in the MO@PG composite, metal oxide embedded in a holey graphene composite retains structural pore stability after a lengthy electrochemical test (after 4000 cycles).

## 2.3. 3D Structured Carbon/Metal-Oxide Composite

The specific capacity, energy, and power density need to be maximized by improving the quality and quantity of the active materials embedded in an electrochemical energy storage device [44]. In this respect, it is important to load as much active material as possible in the electrode to enhance the capacitance, energy, and power in the device [52]. Hence, there were numerous studies conducted to load a large amount of active material into an electrode of an energy storage device without deteriorating the electrochemical performance (specific capacity, rate capability, energy density, and power density) [27]. Specifically, a significant amount of research on composites of porous metal oxide with 3D structured carbon was undertaken to address these challenges.

The characteristics of 3D porous carbon (tunable pore size, wall thickness, high pore volume and surface area, and inner connection among the pore channel) make them extremely suitable for use as a matrix of an active material [27]. As summarized in Figure 10, hierarchical porous carbon, which has macroporous, mesoporous, and microporous properties, can be used to improve the electrochemical performance of interconnected pores of different sizes in an energy storage device. In a composite of metal oxide and 3D carbon, 3D structured porous carbon plays an important role because it forms a hierarchical structure composed of different sized pores (macropores, mesopores, and micropores). Such a hierarchical structure of different sized pores can improve the electrochemical performance of an energy device through unique actions with multiple functions including ion-buffering reservoirs, short ion pathways, ion confinement, and the effective suppression of the volume change during electrochemical reactions. Because metal-oxide nanomaterials are well embedded in the pore structure of a composite, interconnected pores can effectively facilitate the electrolyte penetration and ion diffusion, leading to an enhanced structural stability [63]. In addition, the enlarged surface area of a composite through a porous structure widens the active site by creating numerous contacts with the electrolytic solution, enabling the loading of a large amount of metal oxide to realize a high energy and power density [64].

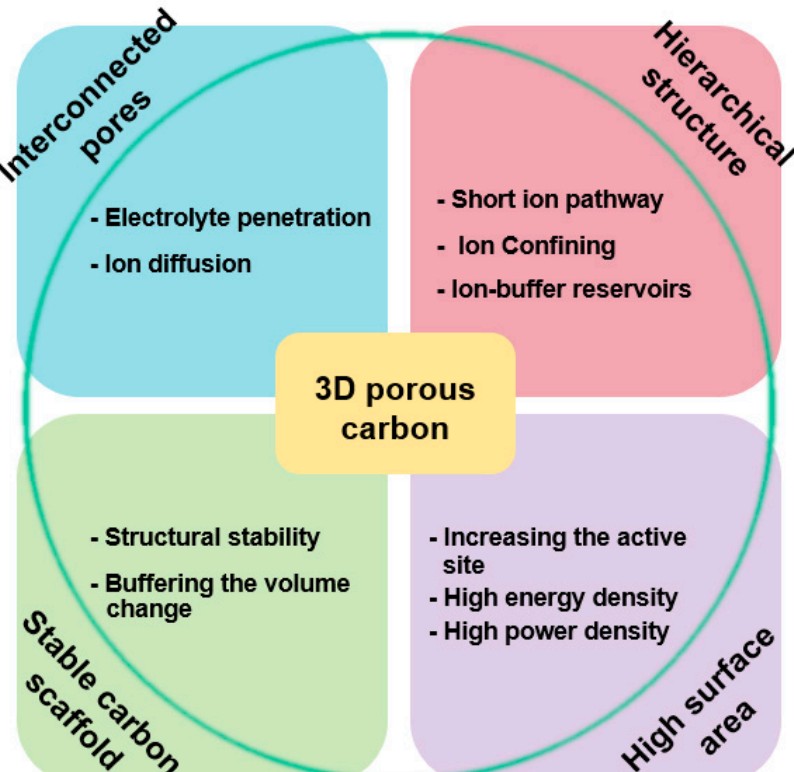

**Figure 10.** Structural properties of three-dimensional (3D) porous carbon and improved properties as an electrochemical energy device electrode of a 3D porous carbon/metal-oxide composite.

For instance, Wang's group reported the use of hierarchical porous graphitic carbon as an electrode material for electrochemical capacitors (Figure 11) [65]. The as-formed porous carbon exhibits a high porosity (high surface area and pore volume), suggesting its use as an appropriate electrode material for electrochemical energy devices. Specifically, the Brunauer–Emmett–Teller (BET) surface area, pore volume, and average pore diameter of porous carbon were measured to be 970 $m^2$/g, 0.69 $cm^3$/g, and 2.85 nm, respectively. In addition, the micropore volume and its ratio (micro volume in the total pore volume) were calculated as 0.3 $cm^3$/g and 0.43, respectively. More specifically, the nitrogen isotherm curves (Figure 11a) display a combined formation of type I in microporous materials and type II in macroporous materials, whereas hysteresis loops are well characterized as mesoporous materials. Within the pore size distribution (Figure 11b), three types of pores, namely micropores (1–2 nm), mesopores (5–50 nm), and macropores (60–100 nm), were observed, suggesting the hierarchical structure of porous carbon. As shown in Figure 11c, this hierarchical structure of 3D porous carbon can enhance the performance of electrochemical energy storage via the synergic functions of various types of pores (micro-, meso-, macropores) in different sizes. Macropores, which have a diameter of 1 μm in the composite, can extend into the particle to form ion-buffering reservoirs. Furthermore, the wall thickness of the porous graphitic carbon around the core is estimated to be less than 100 nm, indicating reduction of the diffusion distance of the electrolyte by this thin wall. The mesopore structures provide a short ion transport pathway through the wall. The micropore structures suggest facile confinement of the ions.

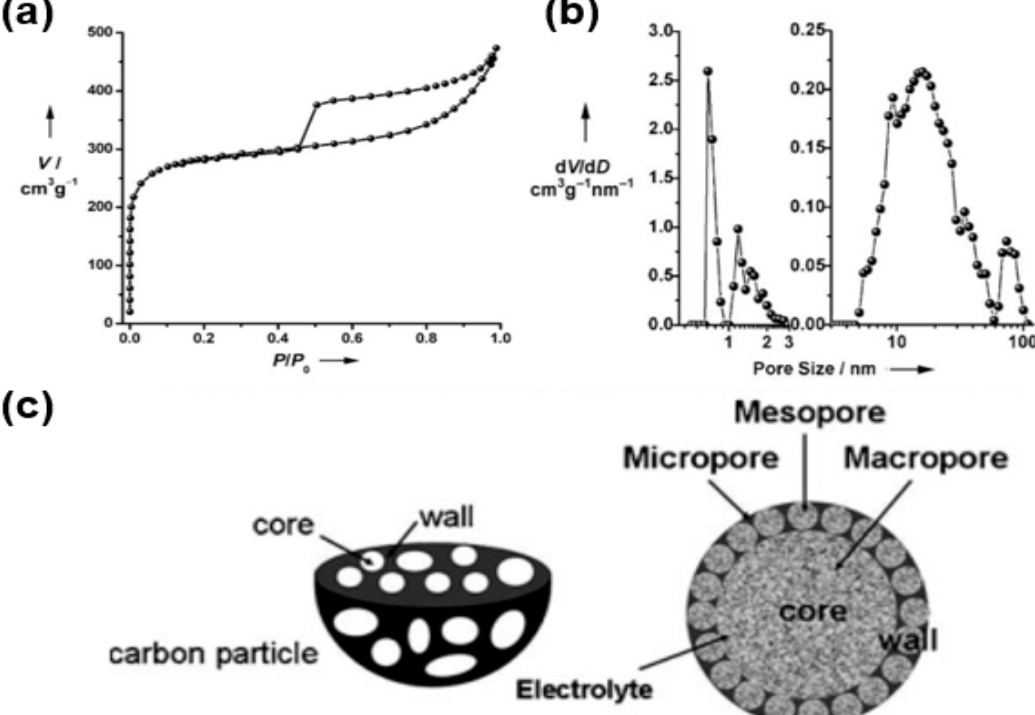

**Figure 11.** (**a**) Nitrogen adsorption–desorption isotherm of hierarchical porous graphitic carbon. (**b**) Pore size distribution of hierarchical porous graphitic carbon. (**c**) Hierarchical porous carbon textures with interconnected pores of macropores, mesopores, and micropores. Reprinted with permission from Reference [65]; copyright 2008 Wiley-VCH.

Three-dimensional porous carbon can act as an extremely good matrix for a composite with metal oxide for use as an electrode material [29]. For instance, Cao et al. synthesized an MnO/C composite with nanoparticles uniformly embedded in a porous carbon matrix to be used as an anode of a lithium-ion battery [66]. As shown in the scheme (Figure 12a) of the energy storage characteristic of the MnO/C composite, this structure facilitates the penetration of an electrolyte through an open

space between internally connected pores and the aligned layers, as well as the effect of shortening the diffusion path of the Li ions during an electrochemical reaction. A high-magnification SEM image (Figure 12b) reveals a layered structure of the composite with a lotus-root-like hollow interior structure based on the distinct contrast between the outer wall and the middle cavity region. In addition, the MnO$_2$ metal oxide in the carbon matrix effectively reduces the mechanical stress caused by a volume change occurring from the lithiation/delithiation during the charging/discharging process. Such an advantageous structural feature leads to a reduced resistance, as shown in the electrochemical impedance spectroscopy (EIS) results (Figure 12c). Specifically, the EIS results of the MnO/C–1.0% PVA composite after 250 cycles at 0.75 A/g show that the charge transfer (R$_{ct}$) and Warbug impedance (Z$_w$) have lower values compared to those of pristine MnO. These results indicate that the 3D porous structure of a MnO/C composite improves the diffusion of lithium ions and electrons compared to pure MnO, suggesting an improved electrochemical performance in energy storage devices. Similarly, Lu et al. synthesized MnO nanoparticles with N-doped carbon coatings to improve the cycling performance and rate capability of such a device [67].

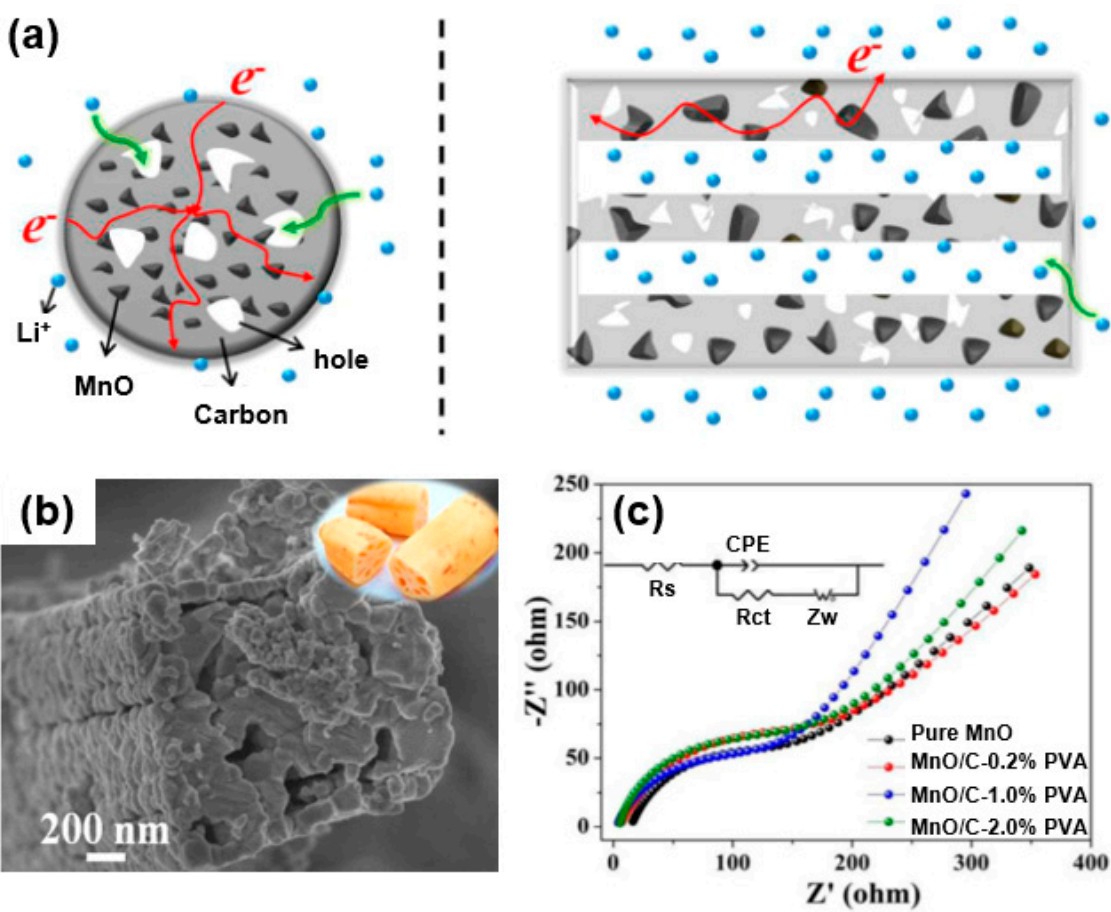

**Figure 12.** (**a**) Energy storage characteristics of MnO/C hybrids: (left) cross-sectional view and (right) axial side cutaway view. (**b**) High-magnification SEM image of MnO/C composite. (**c**) Nyquist plot of MnO/C composite after 250 cycles at 0.75 A/g. Reprinted with permission from Reference [66]; copyright 2017 American Chemical Society.

Another advantage of a composite of 3D structured porous carbon and metal oxide as an electrode of an energy storage device is the effective alleviation of the mechanical expansion and contraction of the metal oxide during the electrochemical reactions because of the hierarchical pore structure of 3D structured carbon. Such a hierarchical structure can enhance the performance of the electrode (cycle stability) [68].

For instance, Zhang et al. showed a composite ($Fe_3O_4$@C) in which $Fe_3O_4$ was confined in a nanoporous carbon framework as an anode of a lithium-ion battery through a self-assembly of Fe $(NO_3)_3$/resol/F127 [69]. SEM and TEM images (Figure 13a–f) show different morphologies of $Fe_3O_4$@C-1 (Figure 13a,b), $Fe_3O_4$@C-2 (Figure 13c,d), and $Fe_3O_4$@C-3 (Figure 13e,f) depending on the amount of $Fe_3O_4$. As shown in the nitrogen sorption isotherm curve (Figure 13g) and pore structure analyses (Figure 13h), it was found that the pore structure of $Fe_3O_4$@C depends on the iron precursor (Fe $(NO_3)_3$/resol/F127) content in the self-assembly process. Specifically, in the case of $Fe_3O_4$@C-3, a large amount of $Fe_3O_4$ loaded into the pores results in a decreased pore volume and surface area. In the case of $Fe_3O_4$@C-2, a proper amount of $Fe_3O_4$ was confined to the porous carbon and showed the largest surface area and pore volume. These composites enable facile contact of the active material with the electrolyte and an effective inhibition of the volume change generated during the lithiation/delithiation process. They also prevent the $Fe_3O_4$ from aggregating or separating from the collector, enabling a large amount of $Fe_3O_4$ loading into the carbon framework with an increased pore volume and surface area owing to the porous structure [70].

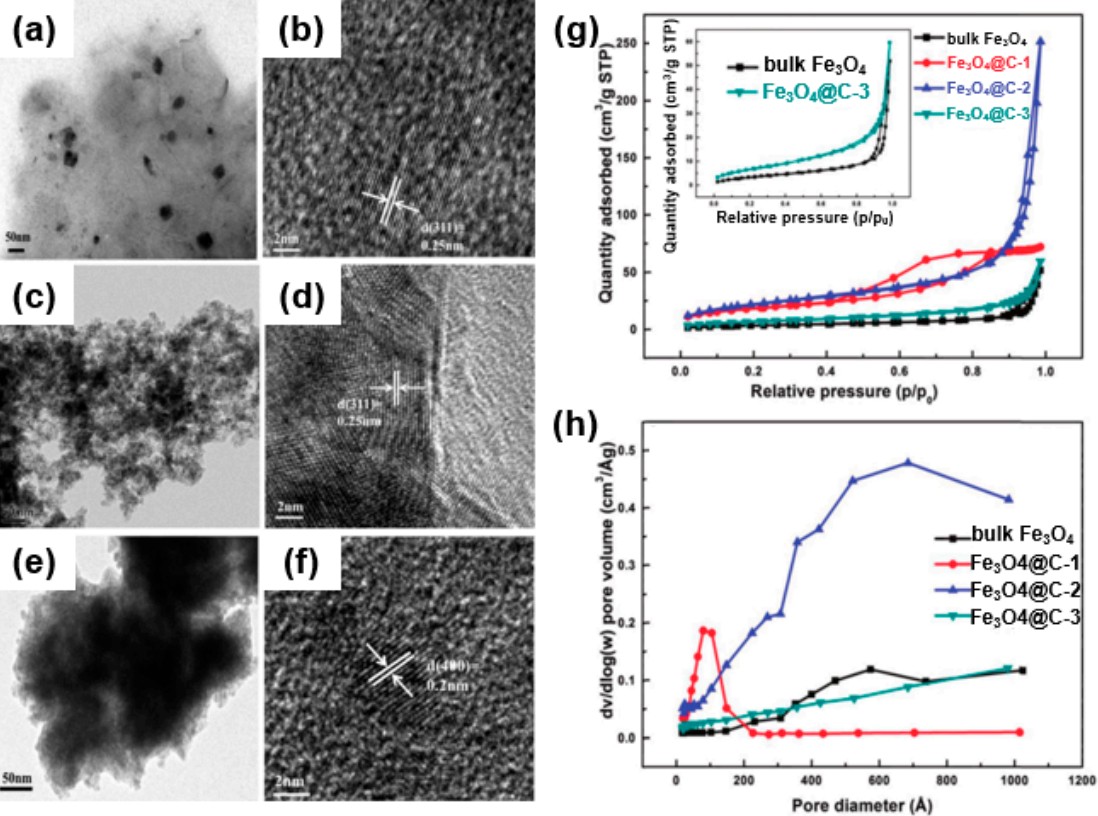

**Figure 13.** TEM images of $Fe_3O_4$ confined in various nanoporous carbon frameworks: (**a,b**) $Fe_3O_4$@C-1, (**c,d**) $Fe_3O_4$@C-2, and (**e,f**) $Fe_3O_4$@C-3. (**g**) Nitrogen adsorption–desorption isotherms for $Fe_3O_4$ and $Fe_3O_4$@C. (**h**) Pore size distribution. Reprinted with permission from Reference [69]; copyright 2015 Royal Society of Chemistry.

In a similar vein, Han et al. synthesized $SnO_2$@CMK-3 composites in the form of ultrafine $SnO_2$ particles encapsulated in tubular mesoporous carbon as electrodes for lithium-ion batteries (Figure 14) [71]. Figure 14a,b show TEM images of a $SnO_2$@CMK-3 composite before and after 100 cycles. As revealed in the TEM figures, the ordered mesostructure was observed to be in a well-maintained form in both samples, whereas the nanoparticles ($SnO_2$) were properly and uniformly encapsulated in the carbon matrix without aggregation. Owing to the high porosity of a porous carbon matrix, it is

possible to load large amounts of SnO₂ nanoparticles into the pores, which effectively accommodates the volumetric change of SnO₂ during lithiation (Figure 14c).

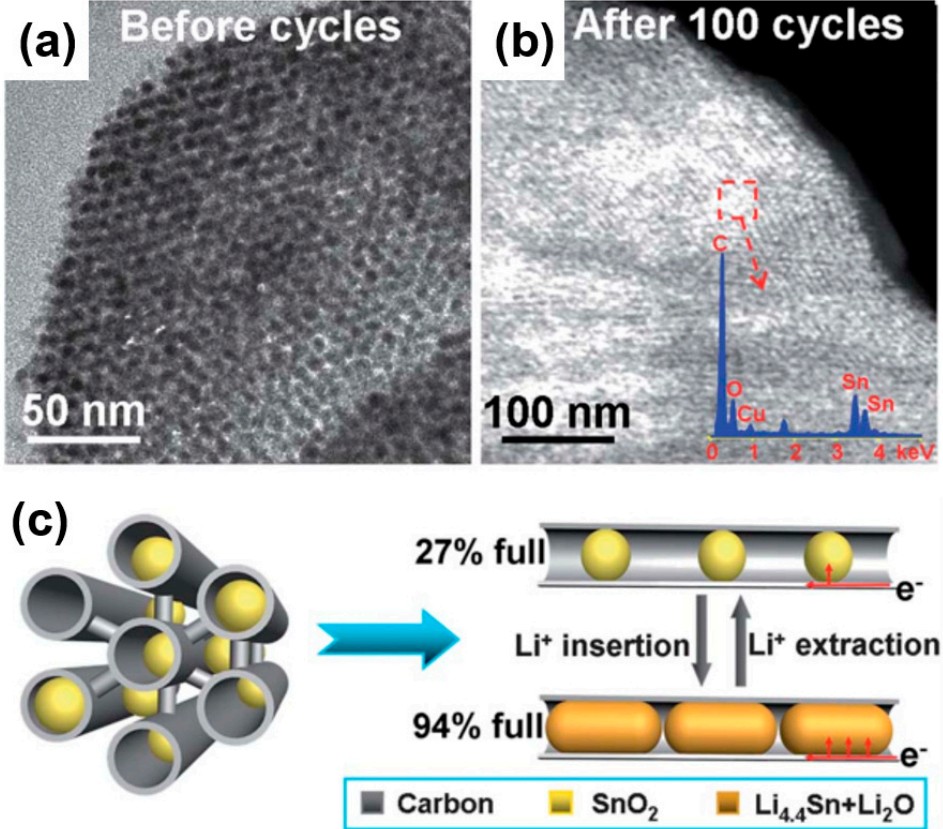

**Figure 14.** (**a**) TEM images of ultrafine SnO₂ particles encapsulated in tubular mesoporous carbon (SnO₂@CMK-3) before cycling. (**b**) Scanning TEM (STEM) images of SnO₂@CMK-3 after 100 cycles at 200 mA/g. (**c**) Adequate space of the mesopore channels acting as a "buffer zone" for accommodating the volume expansion and preventing the pulverization of SnO₂ nanoparticles during the charging/discharging process. Reprinted with permission from Reference [71]; copyright 2012 Royal Society of Chemistry.

Thus, 3D porous carbon with a high surface area, pore volume, and an internally well-connected hierarchical structure allows a large amount of loading of capacitive materials through the formation of a composite with metal oxide in an energy storage device. In addition, enabling an efficient contact with the electrolytes and structural stability during the cycling effectively facilitates the diffusion of lithium ions and electrons, resulting in an improved electrochemical performance.

## 3. Applications of Carbon/Metal-Oxide Composite Materials

### *3.1. Batteries*

#### 3.1.1. Lithium-Ion Batteries

Since the introduction of the first commercial lithium-ion batteries by Sony, numerous studies were conducted to improve the performance of electrochemical energy devices. Currently, the use of lithium-ion batteries is becoming increasingly widespread owing to their many advantages as energy storage devices, including a high capacity, high energy density, rate-response characteristics, and long cycle life.

Lithium-ion batteries (LIB) consist of four parts: anodes, cathodes, separators, and electrolytes. Li-ion batteries, which are currently in commercial use, are composed of LiCoO₂, LiMn₂O₄, and the

like, in which a positive electrode is inserted with lithium, and graphite is used as a negative electrode. Among these options, graphite (anode) forms $LiC_6$ and has a theoretical capacity of 372 mAh/g during the charge/discharge of a lithium-ion battery [72]. However, because of its low theoretical capacity as an energy storage device, there were many studies conducted on metal oxides ($Fe_2O_3$, $SnO_2$, $Fe_3O_4$, $Co_2O_3$, etc.) with a high theoretical capacity. The conversion reaction of metal oxides through the lithiation/delithiation of lithium ions undergoes the following reaction (Equation (2)) [23]:

$$MO_x + 2xe^- + 2x Li^+ \leftrightarrow M^0 + x Li_2O \tag{2}$$

Despite the high theoretical capacity of metal oxide originating from the conversion reactions during the lithiation/delithiation, the metal-oxide-based LIB suffers from a low electric conductivity and low Coulombic efficiency, as well as a rapid capacity reduction caused by a large volume expansion/reduction of the active material. As mentioned in the previous section, new approaches addressed the above problems. As a representative method, carbon-based composites are currently prevailing in which a carbonaceous material is used as a matrix to form a composite with metal oxide. Such an approach can result in a high electrical conductivity of the carbon material, enabling a synergistic effect of the high theoretical capacity of the metal oxide.

As described in the previous section, metal-oxide/carbon composites, including 1D, 2D, and 3D structured composites, were intensively studied through various synthetic approaches.

Firstly, among the various types of carbon/metal-oxide composites, 1D structured CNT/metal-oxide-based composites [73] or carbon-fiber/metal-oxide-based composites [74,75] formed by bonding with a metal oxide were intensively introduced for use as anodes for energy devices (LIBs). Typically, CNTs and networks serve as a buffer layer to accommodate the mechanical strain caused by large volume changes of metal oxides, and they can act as a conductive path (electronic highway) to improve the electrical conductivity of the composite [76]. CNTs also prevent the aggregation of nano-sized metal oxides dispersed on the composites.

For instance, Luo et al. presented the performance of LIB cathodes using a composite of $Mn_3O_4$ mingled with well-aligned carbon nanotubes (SACNTs) [77]. Figure 15a shows that the SACNT film serves as a conduction path for electron migration and acts as a matrix to ensure the adequate dispersion of $Mn_3O_4$ nanoparticles. As displayed in the TEM image (Figure 15b), as-formed composites contain 4 nm of $Mn_3O_4$ nanoparticles, whereas the $Mn_3O_4$/SACNT composite exhibits a capacity of ~342 mAh/g at a high current density of 10 C (Figure 15c). Note that, as an advantageous feature, the 1D structured composite as a free-standing film does not require a conductive agent (e.g., super-P), a binder, or a cathode substrate. In a similar approach, Zhou et al. synthesized a composite of single-walled carbon nanotubes (SWCNTs) with $Fe_2O_3$. The composite exhibits a high capacity (1243 mAh/g) at 50 mA/g, owing to a mitigated lithium-ion path toward the dispersed $Fe_2O_3$ nanoparticles in the SWCNT network. In addition, the alleviation of the volume change of the composite occurring during the delithiation process results in an elongated capacity retention at an increased energy density for long cycles. Similarly, Li et al. used multi-walled CNTs (MWCNTs) as a matrix for anchoring $Fe_3O_4$ nanoparticles to improve the electrical conductivity and structural stability [78].

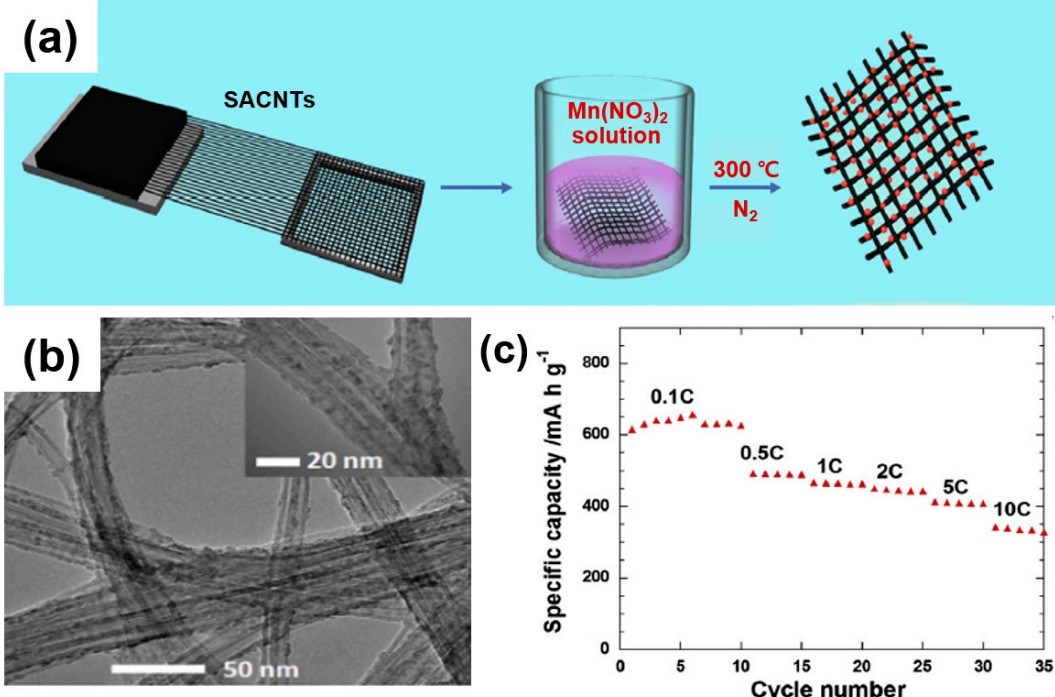

**Figure 15.** (**a**) Schematic illustration of the fabrication of the $Mn_3O_4$/SACNT composite. (**b**) TEM image of the SACNT/$Mn_3O_4$ composites containing 29 wt.% $Mn_3O_4$. (**c**) Rate capability of the $Mn_3O_4$/SACNT composite at various current densities. Reprinted with permission from Reference [77]; copyright 2013 Elsevier.

Metal oxides composited with a 2D structured carbon (graphene, graphene oxide, etc.) were intensively studied as energy storage materials owing to their many structural advantages, leading to an improved LIB performance. More specifically, it is logical to expect an improved LIB performance for a 2D structured carbon composite because of the unique material characteristics of graphene, including high electrical conductivity, excellent mechanical properties, and wide specific surface area [53].

In this regard, Li et al. synthesized a composite of $Co_3O_4$ and graphene, exhibiting a theoretical capacity of 890 mAh/g (approximately twice the theoretical capacity of graphite) through a hydrothermal synthesis followed by an annealing process [79]. The as-formed composite shows a multifunctional structure of 1D $Co_3O_4$ nanowires grown on a 2D graphene membrane. Note that a unique composite structure was prepared through the sequential construction processes where the graphene membrane serves as a conductive substrate and a $Co_3O_4$ nanowall is grown on a graphene nanosheet. Such a composite-based electrode eliminates the need for a copper current collector and a binder to increase the energy density of the battery, leading to an enhanced energy density of the active materials (Figure 16a). As confirmed through SEM images (Figure 16b), $Co_3O_4$ nanowires are grown on a graphene membrane. In the elongated cycle test, the composite showed good stability (Coulomb efficiency of ~100% during 500 charge/discharge cycles and a capacity of 600 mAh/g or higher (Figure 16c).

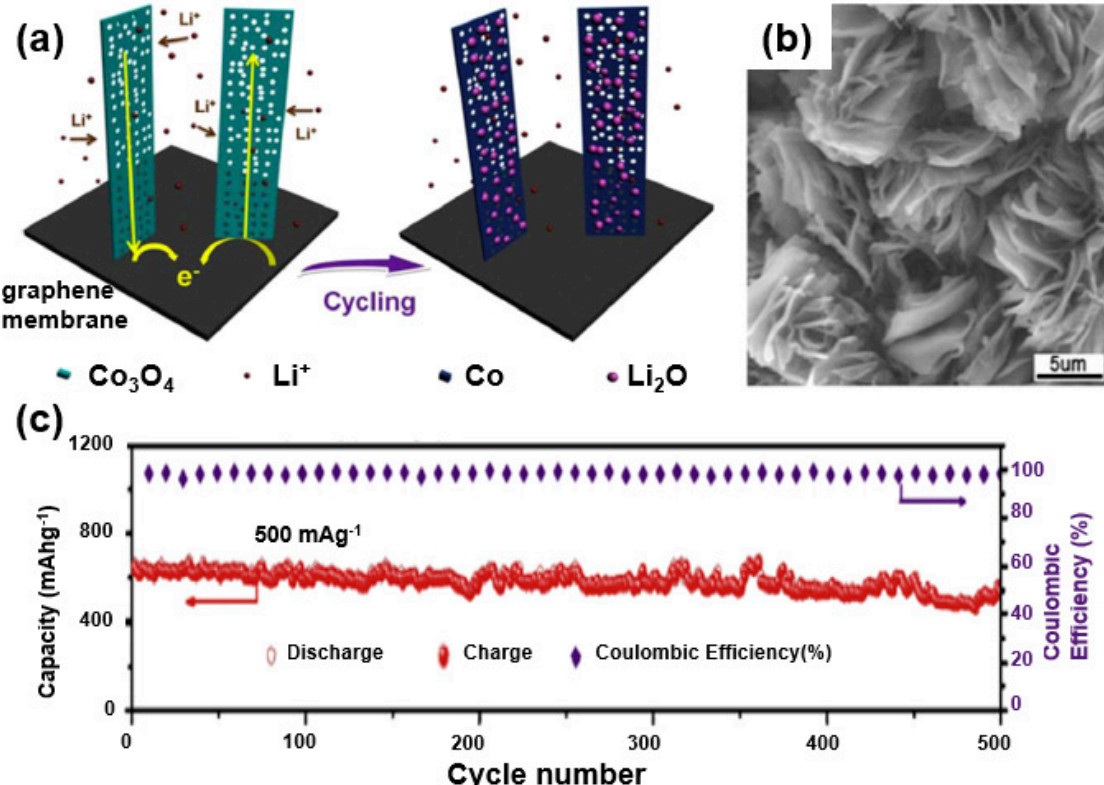

**Figure 16.** (**a**) A schematic of the electrochemical reaction of a $Co_3O_4$ nanowall/graphene membrane composite. (**b**) SEM image of $Co_3O_4$ nanowall/graphene membrane composite. (**c**) Long-term cycling performance of $Co_3O_4$ nanowall/graphene membrane composite. Reprinted with permission from Reference [79]; copyright 2014 Elsevier.

In another approach, Hu et al. published a study in which reduced graphene oxide (rGO) was surrounded by $Co_3O_4$ porous nanofibers to form a composite [80]. The porous nanofiber encapsulates the $Co_3O_4$ nanoparticles and the interconnected graphene oxide sheet on the interconnected formation, thereby suppressing the large volume change of the $Co_3O_4$ particles generated during the battery charging/discharging process, as well as accelerating the movement of electrons and lithium ions. Based on the structure of this composite, a dose of ~900 mAh/g was reported at a current density of 1 A/g and a capacity of ~600 mAh/g at a high current density of 5 A/g.

In addition, some studies demonstrated an improve battery performance by forming a composite in which metal-oxide nanoparticles are placed between the reduced graphene oxide sheets [81] or through a formation on a graphene sheet in the form of metal-oxide nanoparticles and nanorods [82,83]. In addition to the use of graphene-like carbon materials, a study was conducted on the application of carbon sheets as a matrix of a metal-oxide composite. Fu et al. used carbon sheets with glycine as a carbon precursor to make a composite with metal oxide [84]. Chromium (II) oxide nanoparticles are anchored onto the carbon sheets, enhancing the electrochemical performance.

Among these carbon-based composites formed using metal oxides, a 3D carbon-based composite with a porous structure is formed to enlarge the specific surface area, thereby improving the contact area with the electrolyte and facilitating the electron transfer. To improve the lithium-ion battery performance, the 3D carbon network serves as a connection path through which electrons move, and it acts as a buffer for the large volume change of the metal oxide generated during the battery charging/discharging process. Wang et al. improved the mechanical properties and ionic and electrical conductivities of the composites by synthesizing the composites in the form of MnO dispersed in a 3D porous carbon network, and obtained a capacity of 560.2 mA/g at a high current density of 4 A/g [85].

Han et al. reported a composite of SnO$_2$ nanoparticles embedded in ordered mesoporous carbon (CMK-3 or CMK-5) as a carbon matrix [71]. A large amount of SnO$_2$ nanoparticles were added to CMK-5, which has a tubular mesoporous channel with a large pore volume and a thin carbon wall, forming a composite that improves the battery performance. Compared to CMK-5, the relatively thick carbon-walled CMK-3 is unable to effectively trap SnO$_2$ in the mesoporous channel, leading to a relatively lower battery performance (Figure 17a). As confirmed through battery tests, SnO$_2$-80@CMK-5 with an increased loading of SnO$_2$ showed a capacity of 1039 mAh/g after 100 cycles at 200 mA/g (Figure 17b). As shown in the TEM image (Figure 17c), an even distribution of SnO$_2$ nanoparticles with a hexagonal structure was observed among the carbon matrices. Such a unique composite structure of SnO$_2$ nanoparticles uniformly dispersed in the mesoporous channel of CMK-5 can effectively alleviate the large volume changes in the mesoporous carbon wall of CMK-5 during the charge/discharge processes, leading to an enhanced electrochemical performance (Figure 17c).

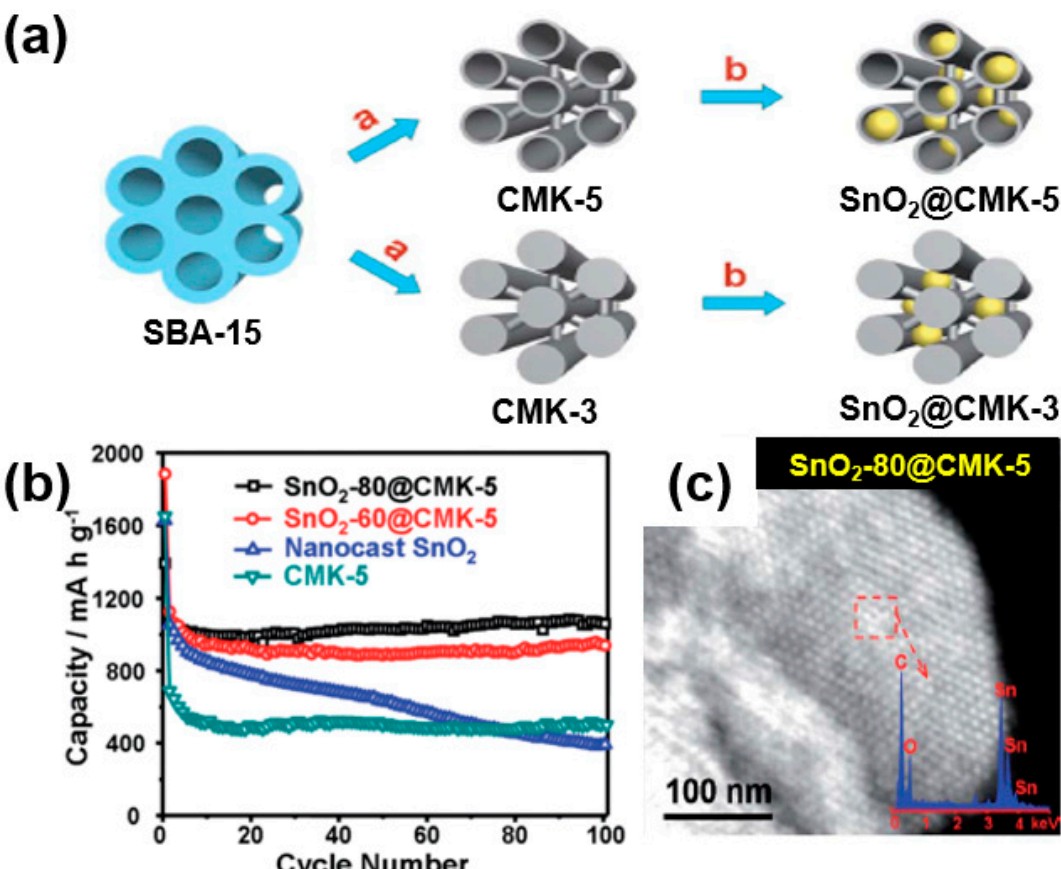

**Figure 17.** (**a**) Schematic illustration of fabrication of composites of SnO$_2$ nanoparticles embedded in ordered mesoporous carbon (SnO$_2$@CMK-5 and CMK-3). (**b**) Cycling performance of SnO$_2$-80@CMK-5, SnO$_2$-60@CMK-5, nano-casted SnO$_2$, and CMK-5. (**c**) STEM image of SnO$_2$-80@CMK-5 composite; the inset shows STEM energy-dispersive X-ray spectroscopy (EDS) analysis. Reprinted with permission from Reference [71]; copyright 2012 Royal Society of Chemistry.

### 3.1.2. Sodium-Ion Batteries

As discussed in the previous section, although LIBs can achieve a high energy density, lithium, an essential element of an LIB, is quite expensive [86]. In this regard, sodium (Na)-ion batteries (SIB) recently gained attention owing to the abundance and lower cost of sodium compared to lithium. However, sodium-based batteries have several disadvantages compared to lithium-based batteries, including a lower standard electrochemical potential (sodium, 2.71 V; lithium, 3.04 V), a relatively larger ionic radius (Na$^+$, 1.02 Å; Li$^+$, 0.59 Å), a relatively slow dynamic response, and a smaller energy

density [86,87]. In particular, the almost double ionic radius of sodium results in a much larger volume change of the active material than that of lithium during the charge/discharge process. In this context, there were intensive studies addressing the above issues of SIBs through the formation of composites of metal oxide and carbonaceous materials with a 1D, 2D, or 3D structure.

Firstly, CNTs as 1D structured carbon materials were used as a matrix in which metal-oxide nanoparticles are well adhered/distributed on the surface of the carbon nanotubes. Using CNTs as a fast-moving path for the electrons, the dispersed metal-oxide nanoparticles can quickly cause an electrochemical reaction of sodium ions and electrons. In addition, CNTs can retard the decrease in battery capacity caused by a large volume change of the metal-oxide particles during the charge/discharge processes.

For instance, Wang et al. reported an improved performance of SIBs by adding $SnO_2$ nanoparticles to CNTs into a composite formation [88]. Rahman et al. also studied the application of $Co_3O_4$, which was extensively studied in LIBs, in sodium-ion batteries [89]. A $Co_3O_4$/CNTs composite was formed using a liquid plasma (Figure 18a). SEM and TEM images show that the CNTs are wrapped in a composite of $Co_3O_4$ particles (Figure 18b). The CNTs are wrapped around the $Co_3O_4$ nanoparticles to improve the contact properties of the CNTs by reducing the volumetric change in the active materials during the charge/discharge processes and by increasing the electrical conductivity. It can be seen from the EIS measurement results that the charge transfer resistance of the $Co_3O_4$/CNT composite is relatively smaller than that of $Co_3O_4$, which indicates that the conductivity of the composite battery is improved (Figure 18c). As a result of testing the charge/discharge cycles, it was confirmed that the capacity of the $Co_3O_4$/CNT composite is maintained at 403 mAh/g even after 100 cycles (Figure 18d). The results indicate that the performance of an SIB can be improved through the composite formation of CNTs with metal oxides.

To improve the performance of the SIBs, it is necessary to form a composite with a carbon material that acts as a small nanoparticle-sized metal oxide, creating a passage through which the electrons can properly move. Graphene not only increases the electrical conductivity but also acts as a matrix in which the metal-oxide particles can be uniformly dispersed and bonded. $Fe_2O_3$, one of the metal oxides applicable to sodium-ion batteries, has the following reaction (Equation (3)) [90]:

$$Fe_2O_3 + 6Na^+ + 6e^- \leftrightarrow 2Fe^0 + 3Na_2O \tag{3}$$

Through such a reaction, the Fe nanoparticles are dispersed in the $Na_2O$ matrix and cause a large volume change.

Liu et al. demonstrated an $Fe_2O_3$/rGO composite-based SIB with an improved device performance [91]. Specifically, they prepared a composite of $Fe_2O_3$ nanoparticles homogeneously dispersed on the surface of reduced graphene through a microwave-assisted method, and showed that the capacity of 289 mAh/g was maintained at 50 mA/g after 50 cycles. Such an enhanced performance can be attributed to the reduced graphene, which alleviates the volume change of the $Fe_2O_3$ nanoparticles and compensates for the low electrical conductivity of the metal oxide. In addition, the large specific surface area of the reduced graphene further improves the contact between the sodium ions and the composite, facilitating a charge transfer reaction, resulting in an improved battery performance.

When nano-sized metal-oxide particles such as $SnO_2$, exhibiting an alloying/dealloying reaction, are dispersed on a graphene surface to form a composite, and a charge/discharge process is performed, the reaction is as shown in Figure 19A. The sodiation/desodiation process can be expressed as follows (Equations (4) and (5)):

$$SnO_2 + 4Na \rightarrow Sn + 2Na_2O \tag{4}$$

$$Sn + 3.75Na \leftrightarrow Na_{3.75}Sn (Na_{15}Sn_4) \tag{5}$$

When sodiation occurs, the metal-oxide particles and $Na^+$ alloy with each other, causing the volume to expand and decrease when a dissociation of alloy occurs. The $SnO_2$/rGO composite

maintains a stable solid electrolyte interface (SEI) during this process, resulting in improved battery performance [92].

In a similar approach, Su et al. prepared $SnO_2$@graphene composites using a hydrothermal process for application toward SIBs [93]. SEM and TEM images show the homogeneously dispersed $SnO_2$ nanoparticles on graphene (Figure 19B-a). The performance of the composite was confirmed through battery charge/discharge tests. As a result, the $SnO_2$@graphene composite showed a capacity of 1942 mAh/g during the first cycle and 741 mAh/g during the second cycle (Figure 19B-b). The rapidly decreasing capacity during the second cycle can be seen as irreversible capacity reduction owing to the generation of an SEI. However, a relatively higher performance is shown compared to $SnO_2$ nanoparticles and graphene alone owing to the structural advantage of the composite formation of $SnO_2$ combined with graphene. Based on a rate characteristic test, when the current density was returned to 20 mA/g after reaching a current density of 640 mA/g, it was confirmed that the capacity was reversibly restored, indicating that the graphene composites retain their structure well even under a high current density.

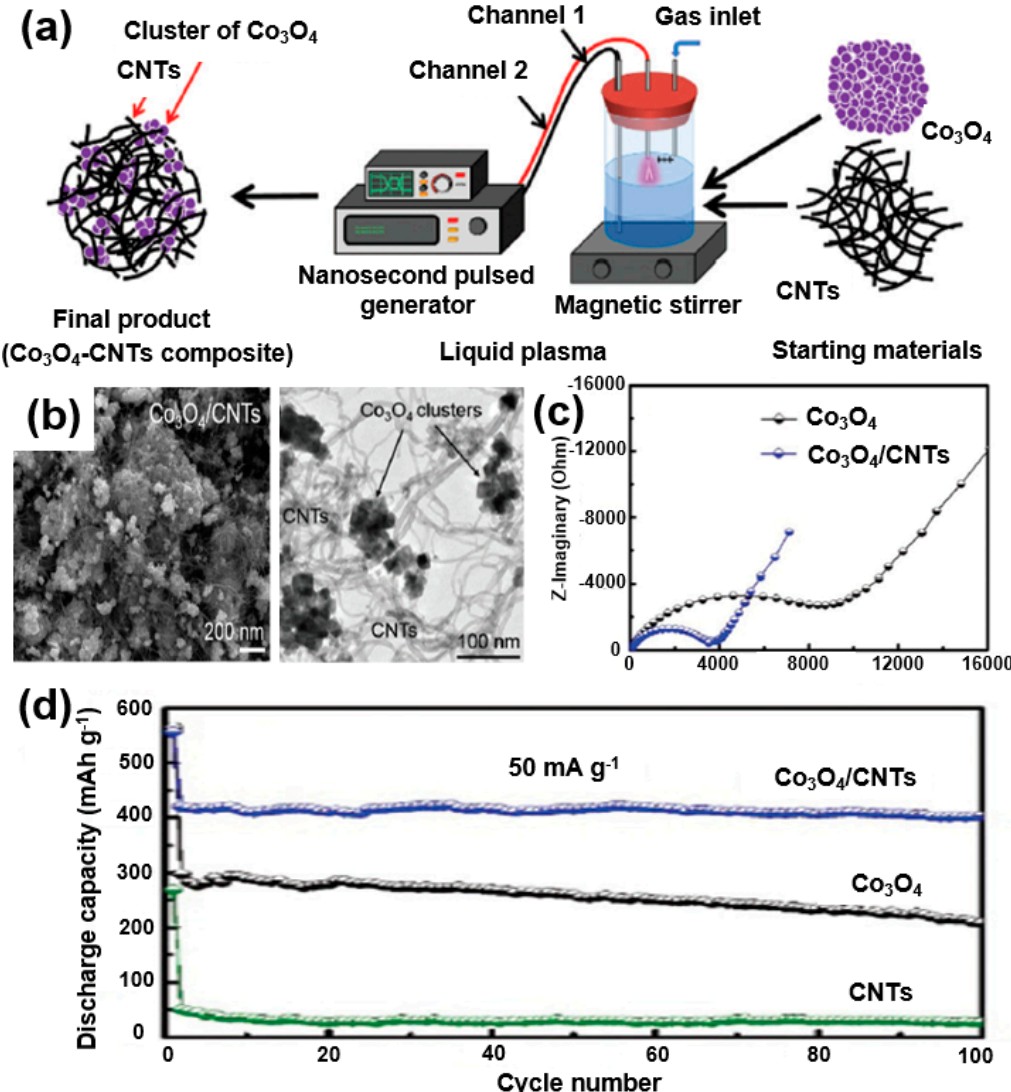

**Figure 18.** (**a**) Schematic illustrations of the synthesis process of $Co_3O_4$/CNT composites. (**b**) SEM and TEM images of $Co_3O_4$/CNT composites. (**c**) Nyquist plots of $Co_3O_4$ and $Co_3O_4$/CNT composites before cycling. (**d**) Cycling performance of $Co_3O_4$/CNTs composites, $Co_3O_4$, and CNTs at a current density of 50 mA/g. Reprinted with permission from Reference [89]; copyright 2015 Royal Society of Chemistry.

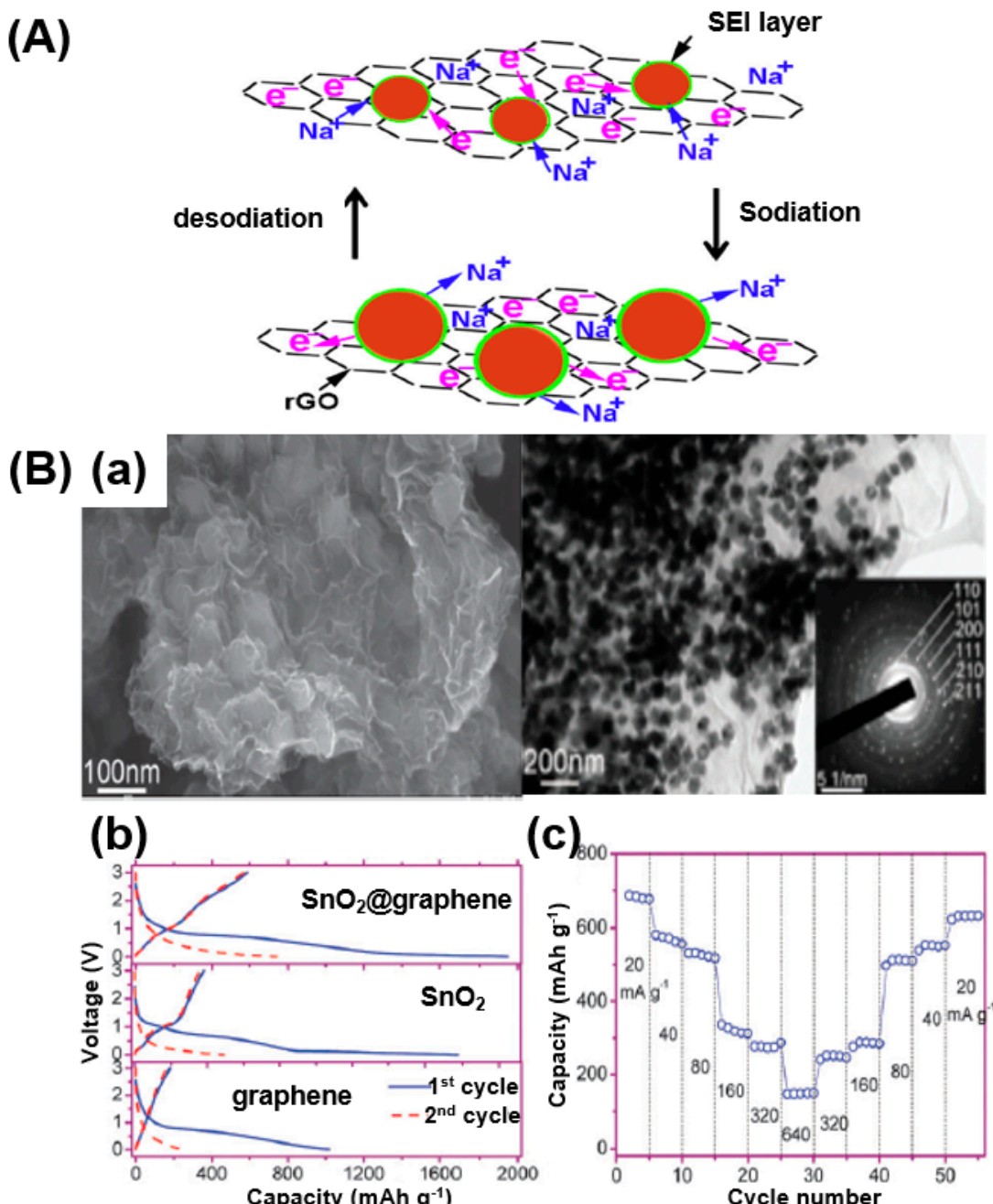

**Figure 19.** (**A**) A schematic illustration of the sodiation/desodiation procedure in an SnO$_2$/reduced GO (rGO) composite. Reprinted with permission from Reference [92]; copyright 2015 Elsevier. (**B**) (**a**) Field-emission (FE)-SEM and TEM images of a SnO$_2$@graphene nanocomposite. (**b**) First and second discharge/charge voltage profiles of SnO$_2$@graphene nanocomposite, SnO$_2$, and graphene. (**c**) Rate capability of SnO$_2$@graphene nanocomposite at different current densities. Reprinted with permission from Reference [93]; copyright 2013 Royal Society of Chemistry.

In the case of a CNT (1D structure) or graphene (2D structure), a metal-oxide composite with short CNTs or graphene with a small size may cause a high contact resistance. To address these issues, Zhao et al. developed SnO$_2$@CEM composites by growing SnO$_2$ nanosheets on a carbonized eggshell membrane (CEM) with a porous structure as a matrix to improve the 1D and 2D structures [94]. Firstly, these 3D hierarchical structures also lead to an enhanced electric conductivity. Using a porous CEM with a unique 3D structure, the composite can be used as a substitute for a copper substrate used in a conventional battery electrode (cathode), leading to an improved energy density of the battery

(Figure 20a). Secondly, because the porous structure facilitates the diffusion of the electrolyte by facilitating the entry and exit of the electrolyte, it can reduce the volume change of the metal oxide, which is a significant problem in SIBs (Figure 20a). An SEM image (Figure 20b) shows that the $SnO_2$ nanosheets grow on the CEM matrix to a size of 300 nm and are in intimate contact with each other. It is anticipated that, when making electrodes with an SnO2@CEM composite, conductive additives such as binders and carbon black may not be needed. Figure 20c shows the cycle stability of the $SnO_2$@CEM composite at 0.1 A/g. Except for the initial irreversible capacity owing to the formation of a solid electrolyte interface, the Coulombic efficiency of the composite was maintained at 92% after the second cycle and reached up to 99% after 100 cycles.

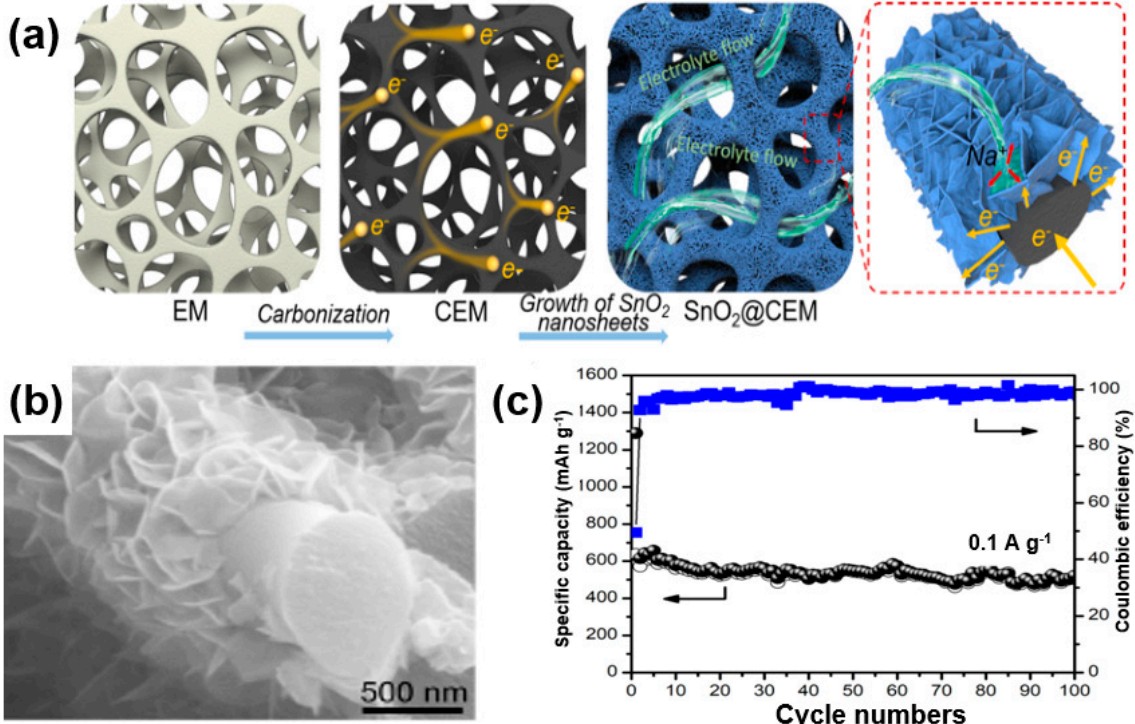

**Figure 20.** (**a**) Schematic illustrations of the synthesis procedure of $SnO_2$ nanosheets on a carbonized eggshell membrane ($SnO_2$@CEM) composite and its structural advantage as an electrode for a sodium-ion battery (SIB). (**b**) SEM image and (**c**) cycle performance of $SnO_2$@CEM composite. Reprinted with permission from Reference [94]; copyright 2018 American Chemical Society.

### 3.2. Supercapacitors

A supercapacitor is used as an energy storage device similar to lithium- or sodium-ion batteries. The power density of a supercapacitor is higher than a battery and, thus, it can be rapidly charged and discharged [95]. However, because the energy density is relatively low, a performance improvement is required. The supercapacitor stores energy through two mechanisms: electrochemical double-layer capacitance (EDLC) and pseudo-capacitance. Currently, commercialized electrodes are made of carbon, and the cyclic stability of a supercapacitor is sufficiently improved owing to the excellent chemical stability of carbon. However, low-capacitance carbon-based supercapacitors generally have an energy density of 3 to 5 Wh/kg, which is a very low energy density (10 to 250 Wh/kg is achieved for a lithium-ion battery) compared to batteries [44]. To increase the capacitance and energy density, a supercapacitor using metal oxide was studied.

When oxidized metal is used alone, the charge transfer and sheet resistances of the supercapacitor electrode increase owing to the low electrical conductivity of the metal oxide (e.g., $10^{-5}$ to $10^{-6}$ S/cm in $MnO_2$) [96]. In addition, a large reduction in capacitance occurs at a high current density. Moreover, like lithium- and sodium-ion batteries, during the charging/discharging process of the supercapacitor,

the volume of an electrode made of metal oxide is changed, thereby decreasing both the contact between the electrodes and the long cycle stability. It is, therefore, necessary to improve the performance of the supercapacitor through synergy by forming a composite of metal oxide and a carbon material. In addition to mitigating the volume change of metal oxide, which is an advantage of carbon, through the formation of a carbon/metal-oxide composite, as well as an improvement in the low electrical conductivity of the electrode, a supercapacitor with a high energy density can be realized.

The high charge/discharge rate of a supercapacitor can be obtained through a short ion movement path, which is an advantage of a 1D structure. For instance, Sankapal et al. developed a ZnO/MWNT composite by growing ZnO on the surfaces of carbon nanotubes (Figure 21a) [97]. Using continuous ion-layer adsorption, ZnO was deposited onto the carbon nanotubes (Figure 21b). A long-term stability test at a current density of 200 mV/s showed a capacity retention rate of close to 83% after 5000 cycles despite the relatively high specific capacitance at 100 F/g (Figure 21c). This is due to the synergistic effects with ZnO based on the improved electrical conductivity, excellent mechanical properties, and short ion transport pathways owing to the 1D structure of carbon nanotubes.

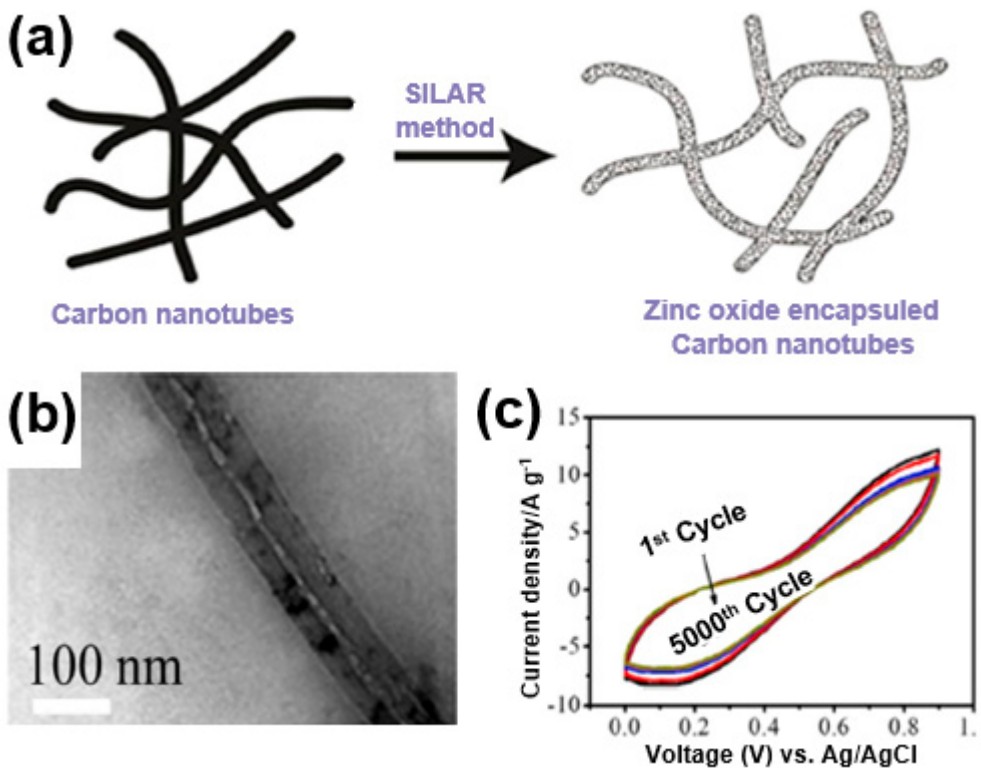

**Figure 21.** (**a**) A schematic illustration and (**b**) TEM image of a ZnO/multi-walled CNT (MWNT) composite. (**c**) Cycle stability performance of ZnO/MWNT composite. Reprinted with permission from Reference [97]; copyright 2016 Elsevier.

A similar approach was used by Yu et al. to study the formation of nanosheet-like NiO particles and composites on carbon nanotubes [98]. NiO nanosheets reduce the ion path and increase the electrochemical active sites through a large specific surface area, enabling a rapid reaction. When NiO nanosheets are present alone, NiO nanosheet particles are dispersed onto carbon nanotubes to prevent the performance degradation of a supercapacitor as the reaction progresses, thereby improving the performance. A high specific capacitance of 1177 F/g was obtained at a current density of 2 A/g.

Studies were conducted in an attempt to achieve application to wearable equipment or small devices by synthesizing a fiber-based supercapacitor with a 1D structure [99,100]. Shi et al. synthesized $MnO_2$@MWCNT composite fibers by twisting MWCNT sheets after injecting amorphous $MnO_2$ onto a sheet of aligned MWCNTs in a solid-state fiber-based supercapacitor, resulting in excellent mechanical

stability, high electrical conductivity, fast ion-diffusion enhanced energy density, rate-limiting properties, and cycle stability of the supercapacitor [100].

Graphene with a 2D structure has a wide specific surface area and excellent electrical conductivity. In addition, graphene oxide and reduced graphene oxide have hydrophilic properties compared to hydrophobic graphene owing to their large number of functional groups on the surface, which is advantageous for synthesizing a composite with metal oxide. In addition, the functional groups present on the surface may act as a redox center of the redox reaction and contribute to a pseudo-capacitance.

Xiang et al. demonstrated $Co_3O_4$/rGO composites prepared using $Co_3O_4$ nanoparticles distributed on reduced graphene oxide through a hydrothermal synthesis [101]. As mentioned earlier, the main pseudo-capacitance comes from the metal oxide. The electrochemical reactions of electrochemically active $Co_3O_4$ are as follows (Equations (6) and (7)):

$$Co_3O_4 + OH^- + H_2O \leftrightarrow 3CoOOH + e^- \tag{6}$$

$$CoOOH + OH^- \leftrightarrow CoO_2 + H_2O + e^- \tag{7}$$

Through these reactions, a highly specific capacitance can be obtained, and the electron transfer between $Co_3O_4$ nanoparticles is accelerated through a reduced graphene matrix, resulting in an energy density of 39.0 Wh/kg and a power density of 8.3 kW/kg. Zhou et al. reported that the supercapacitor performance is improved through the synthesis of $Co_3O_4$ nanoparticles and reduced graphene oxide composites in the form of rolls [102]. Sodium dodecyl sulfate (SDS), a surfactant, and graphene oxide were bonded to each other based on the hydrophilic property. A rolled $Co_3O_4$/rGO composite was synthesized through the ionic bonding of functional groups on the surface of the graphene oxide and $Co^{2+}$ cations (Figure 22a). This unique structure improves the electrical conductivity of the supercapacitor electrodes and tightens the bond between the $Co_3O_4$ and reduced graphene oxide to achieve a 93% specific capacitance retention after 1000 cycles at a scan rate of 20 mV/s, demonstrating a specific capacitance of more than 140 F/g at a current density of 10 A/g and 163.8 F/g at a current density of 1 A/g (Figure 22b,c).

A high energy density, power density, and specific capacitance are essential to improve the performance of a supercapacitor, allowing it to store more capacitance as the loading of the active material increases [103–105].

Li et al. improved the performance of a supercapacitor by forming a composite with $Fe_3O_4$ on hollow porous graphitized carbon with a double shell [103]. The hollow porous carbon increases the specific surface area and electrical conductivity of the metal oxide, and serves as a matrix in which the $Fe_3O_4$ nanoparticles can be uniformly dispersed in the porous carbon (Figure 23a). The composite has a porous formation mixed with micropores and mesopores, and the $Fe_3O_4$ nanoparticles are uniformly dispersed in the graphitized carbon shell (Figure 23b). Owing to the 3D porous structure, the electrolyte penetration of $Fe_3O_4$ nanoparticles can be expected to increase the rate-limiting behavior and cycle stability of the supercapacitor by facilitating a redox reaction at the active sites of the $Fe_3O_4$ nanoparticles. Figure 23c shows a capacitance maintenance rate of 96.7% even after 8000 cycles at a current density of 1.5 A/g, which can be seen in the results obtained through the structural advantages of the hollow porous graphitized carbon/$Fe_3O_4$ composite.

A 3D structure may be formed not only through the use of a porous carbon material, but also by connecting carbon materials with 1D and 2D structures. Dong et al. synthesized graphene/$Co_3O_4$ composites by forming $Co_3O_4$ nanowires through chemical vapor deposition on a 3D porous matrix using graphene foam, and then measured the performance of the supercapacitor [106]. Chen et al. introduced a study on the synthesis of a composite of $MnO_2$ and CNTs in the form of a 3D sponge structure [107].

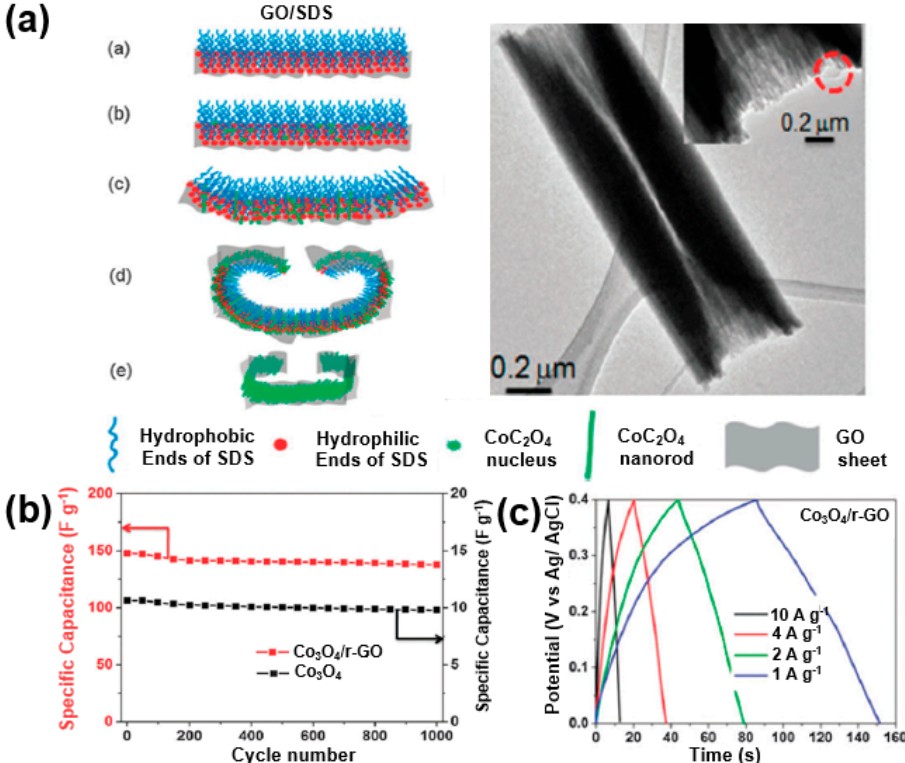

**Figure 22.** (**a**) Schematic illustrations of $CoC_2O_4$ scrolls fabricated in the presence of GO; the inset shows a TEM image of a single $CoC_2O_4$/GO scroll). Adapted with permission from Reference [102]; copyright 2011 Royal Society of Chemistry. (**b**) Specific capacitances of $Co_3O_4$ and $Co_3O_4$/rGO composite. (**c**) Charge/discharge profiles of $Co_3O_4$/rGO composite at various current densities. Reprinted with permission from Reference [102]; copyright 2011 Royal Society of Chemistry.

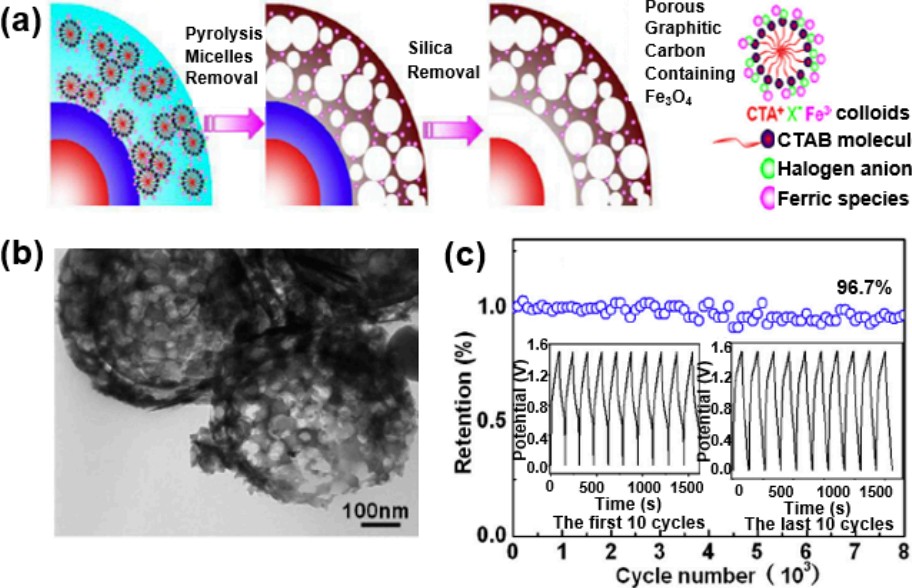

**Figure 23.** (**a**) Schematic illustration of the synthesis of double-shelled carbon (C-C)/$Fe_3O_4$ hollow spheres with a porous structural composite. (**b**) TEM image of the C-C/$Fe_3O_4$ composite. (**c**) Cycle stability of the C-C/$Fe_3O_4$ composite electrode. Reprinted with permission from Reference [103]; copyright 2016 Elsevier.

## 4. Conclusions

This review discussed the use of carbon/metal-oxide composites as electrodes for electrochemical energy devices. Despite the large capacity and high energy density of metal-oxide-based electrochemical energy devices, the low electrical conductivity of the metal oxide and cracks induced through a volumetric change during the electrochemical reactions lead to a deterioration in the electrochemical performances (degradation of the rate capability and cycle stability). This review discussed the solutions to these issues by means of the formation of a carbon/metal-oxide composite, which exhibits synergetic effects and new properties. As described, 1D, 2D, and 3D carbon-based composites bring about the following advantages owing to the unique properties of the composite for each dimension:

1. A 1D carbon nanostructure (e.g., CNT) provides a continuous network for metal oxides with low electrical conductivity, where 1D carbon with a high aspect ratio enables the formation of percolation networks in small quantities. In addition, the carbon in the composite makes it possible to stabilize the metal oxide during cycling owing to its high mechanical robustness.
2. A 2D carbon nanostructure (e.g., graphene) has high electrical conductivity, mechanical strength, and high surface area, being a very good complement to the disadvantages of metal oxides mentioned above. In particular, the metal-oxide/2D-carbon composites of various structures further improve the structural stability of the composite, where 2D carbon with a porous structure enables improved kinetics in electron and ion transport.
3. A 3D carbon nanostructure enables a high loading of the active materials through a high storage capacity (porosity), allowing an increase in the active site for ions during the electrochemical reactions. In addition, the hierarchical structure enables a facilitation of the electrochemical reactions depending on the pore characteristics (e.g., micro/meso/macro) and pore connectivity (or interconnectivity).

Because these different dimensions of carbon have different characteristics owing to the structural characteristics of the material, we focused on describing the effects of composites with metal oxide and carbon for each dimension based on the performance of an electrochemical energy device.

Research on improvement in the performance of electrochemical energy devices is currently underway, and a study on the carbon/metal-oxide composites reviewed in this article is expected to provide an alternative to the energy problems facing mankind.

**Author Contributions:** D.S. and Y.J. contributed equally to this work. H.S. conceived the idea and H.S., D.S., Y.J., performed the basic study and survey. H.S., D.S. and Y.J. wrote the manuscript with the support from K.H. and D.Y.Y. All authors discussed the results and commented on the manuscript.

**Funding:** This research was supported by the National Strategic Project Carbon Upcycling of the National Research Foundation of Korea (NRF) funded by the Ministry of Science and Information and Communications Technology (MSIT), the Ministry of Environment (ME), and the Ministry of Trade, Industry, and Energy (MOTIE) (2017M3D8A2086014).

**Acknowledgments:** We thanks to Sang Wook Kang, Jong-Min Oh, Chulhwan Park, Gun Youl Park, Young Jun Ji, Won Jik Kim and Si Hun Oh owing to their contributions of the preliminary data survey, manuscript preparation and fruitful discussions.

**Conflicts of Interest:** The authors declare no conflicts of interest. The funders had no role in the design of the study; in the collection, analyses, or interpretation of data; in the writing of the manuscript, or in the decision to publish the results.

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
