# Peer review of "Recent Progress of Electrochemical Energy Devices: Metal Oxide–Carbon Nanocomposites as Materials for Next-Generation Chemical Storage for Renewable Energy"

_sustainability, doi:10.3390/su11133694_

Round 1
Reviewer 1 Report
Dohyeong Seok et al have provided a very good and comprehensive review on "Metal Oxide-Carbon Nanocomposite as Materials for Next Generation Chemical Storage for Renewable Energy". The organization of the review into 1D, 2D and 3D carbon materials, also with ample examples from the different applications in electrochemical devices, is quite good. However, the article is very difficult to read/understand due to highly flawed English and much repetitions. I therefore, recommend that the authors improve the language greatly before the paper can be published
Author Response
We appreciate the time and efforts you have dedicated to providing great feedback for our paper. Based on valuable comments, we revised the manuscript by systematically reorganizing and rewriting the texts to remove all flawed English and repetitions. In addition, we further improve the quality of the manuscript after professional English editing. Please see “answer to common comment of all reviewers”.
Answer to common comment of all reviewers
We appreciate all your insightful comments. According to reviewer’s comment, we revised the manuscript for grammar correction by professional English and academic editing services (e.g., Editage). Here we provide with the certificate of English Editing as shown in below.

Reviewer 2 Report
This paper summarizes the recent progress of the metal oxides/carbon composites for energy storage applications. Overall, this article is well organized and can be accepted for publication with a minor revision by improving the quality of the figures.
Author Response
Thank you for your time and great effort in my paper with your insightful comment. We are grateful of your suggestion. We have checked the whole figures in the paper and revise the quality of them : Figure 2, Figure 4, Figure 5, Figure 7, Figure 8, Figure 10, Figure 11, Figure 12, Figure 13, Figure 16, Figure 17, Figure 20, Figure 21, Figure 22, Figure 23. In addition, we further improve the quality of the manuscript after professional English editing. Please see “answer to common comment of all reviewers”.
Answer to common comment of all reviewers
We appreciate all your insightful comments. According to reviewer’s comment, we revised the manuscript for grammar correction by professional English and academic editing services (e.g., Editage). Here we provide with the certificate of English Editing as shown in below.

Reviewer 3 Report
The manuscript “Recent Progress of Electrochemical Energy Devices: Metal Oxide-Carbon Nanocomposite as Materials for Next Generation Chemical Storage for Renewable Energy” summarized the characteristics (chemical, physical, electrical and structural properties) of metal oxide-carbon composites by categorizing the structure of carbon with different dimensions. The applications towards electrochemical energy devices based on the metal oxide/carbon composites were also discussed. The authors have provided sufficient references to back up the conclusions in most cases. However, when reading the manuscript some questions arise, therefore some complementary information and revision should be taken into account before being published in “Sustainability”.
1. In the caption of Figure 7, it is not necessary to add the contents in brackets such as bottom, top, black and so forth, since the legends in the figures have already shown the corresponding samples well. Besides, there are not green spectra in Figure 7g.
2. In the caption of Figure 11, Figure 11e seems more like the low-magnification image of Figure 11f than that of micropores.
3. In the description of Figure 12d-e, it is not appropriate to claim that “charge transfer (Rct) and Warbug impedance (Zw) values were significantly lower in MnO/C composite compared to that of pristine MnO.” In fact, the difference among the EIS plots is not so significant.
4. On page 17, all the images in Figure 13 were assigned to Fe3O4@C-1.
5. In the reaction equation 3-1, the positive symbol of lithium ion was missed.
6. On page 28, it is not appropriate to claim that “conductive additives such as binders and carbon black” since common binders are always not conductive.
7. The references are not homogeneous and sufficient. There are still some other outstanding research works about metal oxide/carbon composites that the authors did not mention, such as: (1) "Li-ion storage performance of MnO nanoparticles coated with nitrogen-doped carbon derived from different carbon sources." Electrochimica Acta 146 (2014): 249-256.; (2) "Nanostructured antimony/carbon composite fibers as anode material for lithium-ion battery." Electrochimica Acta 151 (2015): 214-221.; (3) "Multi-walled carbon nanotubes composited with nanomagnetite for anodes in lithium ion batteries." RSC Advances 5.10 (2015): 7237-7244.; (4) "Chromium (III) oxide carbon nanocomposites lithium-ion battery anodes with enhanced energy conversion performance." Chemical Engineering Journal 277 (2015): 186-193.; (5) "Enhanced electrochemical performances of MoO 2 nanoparticles composited with carbon nanotubes for lithium-ion battery anodes." RSC Advances 5.106 (2015): 87286-87294.; (6) "Carbon composite spun fibers with in situ formed multicomponent nanoparticles for a lithium-ion battery anode with enhanced performance." Journal of Materials Chemistry A 4.25 (2016): 9881-9889.; (7) "Carbon-coated MnO microparticulate porous nanocomposites serving as anode materials with enhanced electrochemical performances." Nano Energy 9 (2014): 41-49. and so on, which are also worthy of discussion.
8. The English writing in manuscript needs to be checked carefully before submission since the meaning of some expressions cannot be understood by the improper use of English or ambiguous description. Especially for capitalization, single and plural, sentence integrity, simple mistakes, and so on.
Author Response
Comment 0:
The manuscript “Recent Progress of Electrochemical Energy Devices: Metal Oxide-Carbon Nanocomposite as Materials for Next Generation Chemical Storage for Renewable Energy” summarized the characteristics (chemical, physical, electrical and structural properties) of metal oxide-carbon composites by categorizing the structure of carbon with different dimensions. The applications towards electrochemical energy devices based on the metal oxide/carbon composites were also discussed. The authors have provided sufficient references to back up the conclusions in most cases. However, when reading the manuscript some questions arise, therefore some complementary information and revision should be taken into account before being published in “Sustainability”.
Answer to comment 0:
Thank you for your comment. Based on valuable comments, we revised the manuscript by providing additional results to the text as well as by addressing technical issues through more elucidated analyses and explanation on the recent results. In addition, we further improve the quality of the manuscript after professional English editing. Please see “answer to common comment of all reviewers”.
Comment 1:
1. In the caption of Figure 7, it is not necessary to add the contents in brackets such as bottom, top, black and so forth, since the legends in the figures have already shown the corresponding samples well. Besides, there are not green spectra in Figure 7g.
Answer to comment 1:
We appreciate for your comments on our manuscript. According to reviewer’s comment, we revised the manuscript by modifying the caption of Figure 7. Specifically, we removed brackets in the caption of Figure 7e and changed the word of explanation of color of the spectra. (green -> purple) Also, we removed two SEM images (b,c, previous) as well.
Comment 2:
2. In the caption of Figure 11, Figure 11e seems more like the low-magnification image of Figure 11f than that of micropores.
Answer to comment 2:
Thank you for your comment. It’s possible to be seemed like the low-magnification image of Figure 11f, more than micropores. According to reviewer’s comment, we revised the manuscript by removing the SEM images of Figure 11c-f to avoid confusion.
Comment 3:
3. In the description of Figure 12d-e, it is not appropriate to claim that “charge transfer (Rct) and Warbug impedance (Zw) values were significantly lower in MnO/C composite compared to that of pristine MnO.” In fact, the difference among the EIS plots is not so significant.
Answer to comment 3:
Thank you for your comment. As reviewer pointed out, it’s not appropriate to claim of charge transfer and Warbug impedance values among MnO/C composite and pristine MnO. Specifically, MnO/C-1.0% PVA composite has lower charge transfer and Warbug impedance value compared to pristine MnO. According to reviewer’s comment, we revised the manuscript by describing Figure 12c (Old Figure 12e) more specifically and comprehensively.
Comment 4:
4. On page 17, all the images in Figure 13 were assigned to Fe3O4@C-1.
Answer to comment 4:
Thank you for your comment. As reviewer pointed out, the text for Figure 13a-f should be revised to show each different composite, such as Fe3O4@C-1, Fe3O4@C-2, Fe3O4@C-3. According to reviewer’s comment, we revised the manuscript by through re-notations.
Comment 5:
5. In the reaction equation 3-1, the positive symbol of lithium ion was missed.
Answer to comment 5:
Thank you for your comment. According to reviewer’s comment, we revised the manuscript by addition of positive symbol of lithium ion in the reaction equation 3-1.
Comment 6:
6. On page 28, it is not appropriate to claim that “conductive additives such as binders and carbon black” since common binders are always not conductive.
Answer to comment 6:
Thank you for your comment. According to reviewer’s comment, we revised the manuscript (text on page 28) by more clarifying non-conductive additives in the composite as follow.
“SnO2@CEM composite may not need conductive additives and binders to make electrodes”.
Comment 7:
7. The references are not homogeneous and sufficient. There are still some other outstanding research works about metal oxide/carbon composites that the authors did not mention, such as: (1) "Li-ion storage performance of MnO nanoparticles coated with nitrogen-doped carbon derived from different carbon sources." Electrochimica Acta 146 (2014): 249-256.; (2) "Nanostructured antimony/carbon composite fibers as anode material for lithium-ion battery." Electrochimica Acta 151 (2015): 214-221.; (3) "Multi-walled carbon nanotubes composited with nanomagnetite for anodes in lithium ion batteries." RSC Advances 5.10 (2015): 7237-7244.; (4) "Chromium (III) oxide carbon nanocomposites lithium-ion battery anodes with enhanced energy conversion performance." Chemical Engineering Journal 277 (2015): 186-193.; (5) "Enhanced electrochemical performances of MoO 2 nanoparticles composited with carbon nanotubes for lithium-ion battery anodes." RSC Advances 5.106 (2015): 87286-87294.; (6) "Carbon composite spun fibers with in situ formed multicomponent nanoparticles for a lithium-ion battery anode with enhanced performance." Journal of Materials Chemistry A 4.25 (2016): 9881-9889.; (7) "Carbon-coated MnO microparticulate porous nanocomposites serving as anode materials with enhanced electrochemical performances." Nano Energy 9 (2014): 41-49. and so on, which are also worthy of discussion.
Answer to comment 7:
Thank you for providing your suggestions. According to reviewer’s comment, we revised the manuscript by citing suggested references in the relevant text part to describe more on metal oxide/carbon composites (ref 63, 67, 74, 75, 76, 78, 84).
Comment 8:
8. The English writing in manuscript needs to be checked carefully before submission since the meaning of some expressions cannot be understood by the improper use of English or ambiguous description. Especially for capitalization, single and plural, sentence integrity, simple mistakes, and so on.
Answer to comment 8:
Thank you for your comment. According to reviewer’s comment, we revised the manuscript for grammar correction by professional English and academic editing services (e.g., Editage). Please see answer to “answer to common comment of all reviewers”.
Answer to common comment of all reviewers
We appreciate all your insightful comments. According to reviewer’s comment, we revised the manuscript for grammar correction by professional English and academic editing services (e.g., Editage). Here we provide with the certificate of English Editing as shown in below.

Round 2
Reviewer 1 Report
The authors have provided an adequate response/corrections to my major concerns, and also the concerns of the referee.
Reviewer 3 Report
All the questions and comments raised by the reviewers were legitimately explained and revised. The accuracy and detail of the manuscript were also improved further after revision.